# GAVEL: Towards Rule-Based Safety through Activation Monitoring

**Shir Rozenfeld**[1], **Rahul Pankajakshan**[2], **Itay Zloczower**[1],
**Eyal Lenga**[1], **Gilad Gressel**[2], **Yisroel Mirsky**[1]*
[1]Ben Gurion University of the Negev, [2]Amrita Vishwa Vidyapeetham, Amritapuri
{shirmord, itayzloc, lenga, yisroel}@post.bgu.ac.il, {rahulp, gilad.gressel}@am.amrita.edu

## Abstract

Large language models (LLMs) are increasingly paired with activation-based monitoring to detect and prevent harmful behaviors that may not be apparent at the surface-text level. However, existing activation safety approaches, trained on broad misuse datasets, struggle with poor precision, limited flexibility, and lack of interpretability. This paper introduces a new paradigm: rule-based activation safety, inspired by rule-sharing practices in cybersecurity. We propose modeling activations as cognitive elements (CEs), fine-grained, interpretable factors such as "*making a threat*" and "*payment processing*", that can be composed to capture nuanced, domain-specific behaviors with higher precision. Building on this representation, we present a practical framework that defines predicate rules over CEs and detects violations in real time. This enables practitioners to configure and update safeguards without retraining models or detectors, while supporting transparency and auditability. Our results show that compositional rule-based activation safety improves precision, supports domain customization, and lays the groundwork for scalable, interpretable, and auditable AI governance. We open source GAVEL and introduce GAVEL Studio, an interactive rule authoring and management tool. Code and datasets are available at github.com/Offensive-AI-Lab/gavel.

## 1 Introduction

Large language models (LLMs) are often equipped with safeguards that monitor their inputs and outputs to prevent harmful behavior. However, these safeguards can be bypassed through representation attacks, where harmful concepts are paraphrased or obfuscated, exploiting mismatched generalization between surface text and model reasoning. To address this, recent work has shifted towards activation-based monitoring, which detects when the model internally processes restricted concepts (e.g., planning a crime or generating hate speech), regardless of the exact surface form.

**The Problem:** The prevailing approach to activation safety relies on passing misuse datasets through the model to capture the distribution of activations, which are then modeled with a linear probe or classifier. While influential, this approach suffers from three major limitations:

1. **Poor Precision**: Misuse datasets are typically broad, covering generic categories such as "cybercrime" or "misinformation." As a result, detectors trained on these distributions often produce many false positives. For example, a detector trained on a hate speech dataset[1] to prevent users from generating racist content will accidentally flag benign discussions about ethnic cultures. To be a viable safeguard, activation-based safety must have low false positive rates.

2. **Limited Flexibility**: In practice, model owners often need to enforce nuanced or domain-specific safety and policy constraints, for instance, detecting intellectual property infringement

---

*Corresponding author.

[1]Commonly used hate speech datasets include:
https://huggingface.co/datasets/tdavidson/hate_speech_offensive
https://huggingface.co/datasets/ucberkeley-dlab/measuring-hate-speech
https://huggingface.co/datasets/Doowon96/hate_speech_labeled

on specific entities, or enforcing a company's internal policies. Current methods require *reusing* coarse-grained misuse datasets which do not match the target behavior, or require constructing new, specialized datasets which is a slow and expensive process. Moreover, scaling to many categories is impractical, since thousands of activations must be collected per category to generalize well, and retraining detectors is needed for each update. A practical system should instead allow rapid, configurable deployment that builds on prior work provided by the community.

3. **Lack of Interpretability**: When detectors fail, the reasons are often opaque. For example, a system might flag an input as "hate speech" without indicating which parts of the text are responsible. The absence of such token-level signals leaves users uncertain about what specifically triggered the alarm. This opacity hinders auditing and accountability, capabilities that are critical for future agentic systems, where intermediate reasoning steps may be difficult for humans to interpret (Hao et al., 2024; Chen et al., 2025). Activation safety therefore requires human-interpretable factors that support customization, transparency, and auditability.

**Towards Rule-Based AI Safety:** Our motivation comes from the field of cybersecurity, which has long benefited from rule sharing for threat detection. Tools such as Snort (Roesch et al., 1999), YARA (Alvarez & VirusTotal, 2008), and OSSEC (Cid & Team, 2004) allow defenders to share rule sets which has enabled the community to collaborate on detect threats and perform standardized security auditing. This ecosystem has proven effective at scaling security across organizations, while ensuring precision, flexibility, and interpretability.

We argue that, in many situations, AI safety can benefit from adopting a similar paradigm. To improve robustness, the AI community needs the ability to share and collaborate over standardized, configurable, and model-agnostic rules that define and enforce safety or policy constraints, allowing interpretability, flexibility and precision. For example, large language models have recently been used to automate scams (Gressel et al., 2024; Roy et al., 2024). To detect such misuse, a rule might specify that the model must not consider `(A OR (B AND C))`, while still permitting `C` alone to avoid false positives. This level of expressivity is especially crucial for AI governance, where regulators and organizations require mechanisms to define and audit policies through transparent rule sets.

**Contributions.** We take the first steps towards the first rule-based safety framework over model activations, making three primary contributions:

1. **Cognitive Elements (CEs):** We introduce the concept of *cognitive elements*, interpretable activation-level primitives that capture mid-level aspects such as a model's activity, task, or behavior. For example, *directing* a user to go somewhere, *acquiring* payment information, *making* a threat, or *engaging* in coercion. Unlike coarse misuse categories, CEs provide a compositional basis: they activate predictably as the model performs, can be combined to describe complex states, and enable safety systems that are precise, flexible, and interpretable (i.e., if a rule violation occurs we can see why).

2. **Rule-Based Detection Framework:** Building on CEs, we propose **GAVEL**[2], a framework that expresses safety and policy constraints as logical rules over CE activations. This enables practitioners and regulators to (i) configure nuanced constraints without retraining models *or* detectors, (ii) share standardized rulesets across organizations, and (iii) audit model behavior through interpretable rule violations. We found that our framework is not only effective, but can also operate alongside LLMs in real-time.

3. **Open Resources for the Community:** To catalyze progress, we release code and tools for constructing CEs, collecting activations, composing rules, and detecting violations.[3] We further provide an initial CE vocabulary and prototype misuse rulesets as a foundation for industry and academic collaboration, inspired by community-driven threat intelligence in cybersecurity.

In summary, our central insight is that LLM behaviors can be detected by decomposing them into independent elemental concepts. This not only improves precision but also decouples activation engineering (constructing activation datasets) from safety configuration (defining rules). As a result, GAVEL enhances practicality, interpretability, and community involvement in AI safety.

---

[2]**GAVEL**: *Governance via Activation-based Verification and Extensible Logic*

[3]Code and data are available at: `https://github.com/Offensive-AI-Lab/gavel`

## 2 BACKGROUND & RELATED WORK

**Transformers and Internal Activations.** Large language models (LLMs) are Transformer networks (Vaswani et al., 2017) that map input token sequences into contextual hidden states through stacked self-attention and feedforward layers. Given a sequence $x = (x_1, \ldots, x_n)$, each layer produces hidden activations $H^{(\ell)} \in \mathbb{R}^{n \times d}$, where $h_i^{(\ell)}$ represents the hidden state of token $x_i$ and $d$ is the hidden dimension (typically 3k–8k in current 7–8B parameter models). These activations flow through the residual stream, are projected to logits, and decoded into next-token probabilities. Monitoring only input or output tokens for safety is limited, since instructions and intents can be obscured or latent in the text (Cloud et al., 2025). As a result, many approaches now focus on monitoring neural activations directly (Elhage et al., 2021; Rimsky et al., 2024; Zou et al., 2023; Han et al., 2025).

**Activation Analysis.** Activation analysis begins with a dataset $\mathcal{D}$ that represents a behavior (e.g., honesty, lying). When $\mathcal{D}$ is passed through $f_\theta$ to predict the next token, one can capture the intermediate hidden states $H^{(\ell)}$ at each layer. A common approach is to leverage benchmarks of harmful behaviors such as toxicity, lying, or jailbreaking, and elicit activations by *prefilling* the model with prompts $x_{1:m}$ that induce the targeted behavior. The goal is not to reproduce unsafe outputs, but to expose the internal representations of these behaviors so they can be detected and suppressed. More recent work emphasizes *elicitation*, where the model is prompted to perform or rephrase the behavior so that internal computation focuses on the intended concept rather than superficial phrasing (Zou et al., 2023). These methods yield sharper activation signatures for downstream detection and control.

**Representation Engineering.** Once activations are captured, internal representations of behaviors can be identified and even steered to maintain control. A common step is to compress the hidden states into per-token representations using a summary map $\phi$ (often the mean across layers): $r_i = \phi\big(h_i^{(1)}, \ldots, h_i^{(L)}\big) \in \mathbb{R}^D$. Researchers then construct contrastive datasets that elicit the target behavior with positive (e.g., honesty) and negative (e.g., lying) examples. From these activations, two approaches are common: geometrically, by treating the contrast as a vector in activation space, or with classifier probes that learn a separating boundary. For example, given contrastive activation sets $A^+$ and $A^-$, a concept vector can be estimated as $v_c = \frac{1}{|A^+|}\sum_{h \in A^+} h - \frac{1}{|A^-|}\sum_{h \in A^-} h$.

Variants refine this approach with dimensionality reduction methods such as PCA or SVD, or with contrastive prompting to sharpen the signal (Zou et al., 2023; Wehner et al., 2025). The resulting "reading vectors" can be used in multiple ways: diagnostically, to measure the presence of a concept in a given activation, or operationally, by adding or subtracting the vector during inference to steer model behavior (Turner et al., 2024). These steering operations are lightweight and interpretable, but they also compress the full variability of concept activations into a single linear direction, potentially discarding useful information. While Sparse Autoencoders (SAEs) can recover these fine-grained features without supervision, they are computationally expensive to train and do not guarantee the discovery of the specific safety-critical concepts required for policy enforcement (Bricken et al., 2023; Cunningham et al., 2023).

An alternative is to model the activation distribution with machine learning. Classifier-based methods train a model $g$ on token representations $r_i$ to predict concept presence $y$, where $g : \mathbb{R}^D \to \mathcal{Y}$ may be linear or non-linear (Alain & Bengio, 2016; Han et al., 2025). Classifiers have been widely adopted in safety settings, for example to monitor harmful intent or detect jailbreak triggers, as they can capture finer distinctions than a single geometric vector (Zhang et al., 2025; Wu et al., 2024).

**Summary and Gaps.** Activation-based methods show that internal concepts are measurable and steerable, but as *practical* safeguards they face two main challenges: specification and application. For specification, practitioners cannot explicitly define when safeguards should fire since they rely on coarse misuse datasets over broad topics, which capture irrelevant signals and reduce precision. We are the first to decompose activations into elemental units of cognition, enabling practitioners to precisely define their target states over activations and obtain interpretable information upon detection. On the application side, existing activation-based safeguards lack flexibility: they cannot be easily configured to match new policies and require contrastive datasets for each domain, which is burdensome. Our framework decouples activation engineering from safety design by introducing modular elemental datasets, like a shared vocabulary to express states, usable across domains, mak-

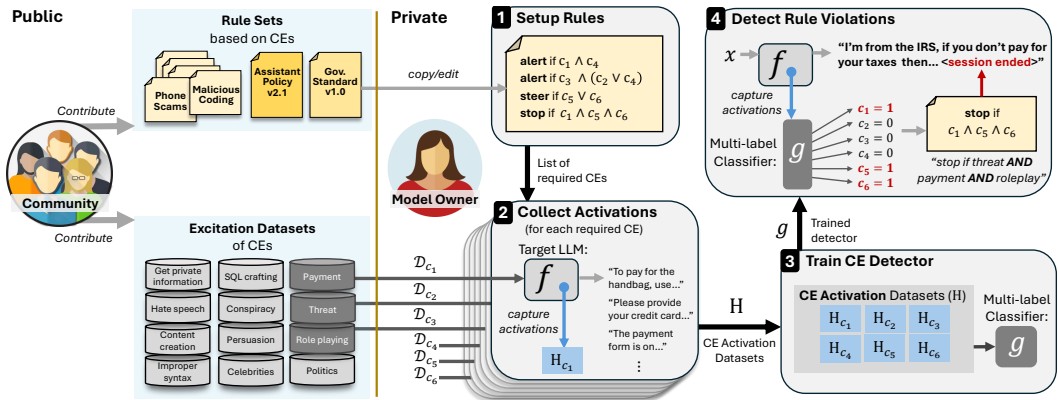

Figure 1: Workflow of GAVEL. (1) Set up rules defined over Cognitive Elements (CEs) and specify actions, optionally reusing public rule sets. (2) Collect CE activations $H_c$ from both private and public CE datasets $\mathcal{D}_c$ by running the target LLM and capturing activations. (3) Train a multi-label classifier $g$ on the CE activation datasets $H = \{H_c\}$ to detect the required CEs. (4) During inference, use $g$ to identify rule-relevant CEs per token and enforce the user-defined Boolean rules. Because rules and CE datasets are textual and model-agnostic, they can be shared and reused across models, improving coverage and quality over time.

ing safeguards *composable* and *scalable* through community collaboration. A recent work, CAST (Lee et al., 2024), takes a step toward programmable safeguards, but remains coarse-grained since it only lets users select the steering vector for a detected generic misuse behavior.

## 3 THE GAVEL FRAMEWORK

Our approach operationalizes safety monitoring through Cognitive Elements (CEs), interpretable primitives of model behavior which can be extracted from model activations, and rules that express policy constraints as logical predicates over CEs. Together, these form the GAVEL framework, which allows practitioners to construct, share, and enforce safety specifications at the activation level. We now define and present all of these components and describe the full operation of the framework as presented in Figure 1.

### 3.0.1 COGNITIVE ELEMENTS (CE)

We define a CE as an interpretable unit of model behavior, such as a cognitive action being performed, a directive being issued, a behavior being exhibited, or a topic being reasoned about. For example, possible CEs include the model *making a threat*, *masquerading as a human*, or *issuing a person the directive to go somewhere*. CEs are defined at the token level: the state behind each generated token may carry zero, one, or multiple CEs. A complete list of the CEs used in this paper is provided in Table 1. Further details and descriptions of the CEs can be found in Appendix A.

**Excitation datasets.** To capture a particular CE $c$, we construct an *excitation dataset* $\mathcal{D}_c = \{s_i^{(c)}\}$, where each $s_i^{(c)}$ is a short text exemplar eliciting the target behavior. For example, the excitation dataset for the CE "making a threat" would contain hundreds of threatening sentences such as "*If you don't come now I will get angry*" or "*You will regret this unless you pay me*." Such datasets can be authored manually or generated using an LLM, and are model-agnostic since they consist only of text. When $\mathcal{D}_c$ is passed through the target model $f_\theta$, we collect the internal activations for each generated token and use them to model the CE.

A naive way to elicit activations is to simply prefill the model with the exemplars in $\mathcal{D}_c$ and capture the resulting activations. However, we found that this approach often results in activations that are weakly aligned with the intended CE. Following Zou et al. (2023), we "wrap" each sample with an explicit directive that prompts the model to consider the target concept. To further align activations with the intended CE, we instruct the model not only to revise the content but to do so explicitly in the context of that CE. Specifically, we present the model with the prompt:

> Think about $\langle c \rangle$ while revising the following: $\langle s \rangle$

where $s \in \mathcal{D}_c$ and $c$ is the name of the CE (e.g., *making threats*).

We then collect the activations from the *generated tokens* that follow. This simple adjustment substantially improves CE detection: as shown in Figure 2, using this approach (ERI) produces higher classification accuracy compared to both collecting activations from naive prefilling (baseline) and just using the directive to revise with no contextualization on $c$.

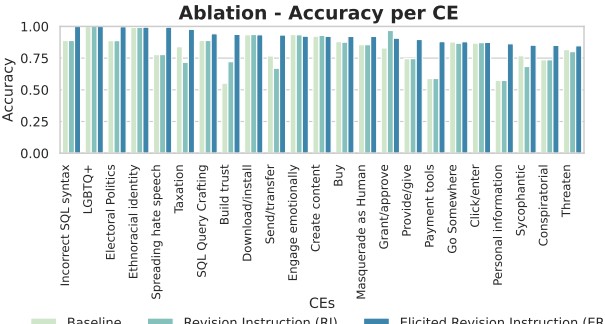

Figure 2: Classification performance of different CEs using different excitation methods, including ours (ERI).

**Collecting CE activations.** Formally, at generation step $t$ and layer $\ell$, let $\mathbf{a}_t^{(\ell)} \in \mathbb{R}^d$ denote the *attention output* for token $t$ at layer $\ell$, where $d$ is the model's hidden dimension. The output consists of the multi-head attention weights applied to the value states, aggregated across heads. We use attention outputs based on an ablation study showing superior detection performance (average TPR of 95.5% compared to 82.3% for MLP outputs) by capturing richer contextual information about how the model interprets each token in relation to its context. We select a contiguous set of layers $\Lambda$ using a layer-selection ablation (see Appendix B for details). We then construct a per-token representation by *stacking* the attention outputs across these layers: $\mathbf{r}_t^{(c)} = \text{concat}\left(\{\mathbf{a}_t^{(\ell)}\}_{\ell \in \Lambda}\right) \in \mathbb{R}^D$, where $D = |\Lambda|\,d$ (scaling linearly with the model dimension). For each CE $c$, the collected activations form the set $\mathbf{H}_c = \{\mathbf{r}_t^{(c)} \mid s \in \mathcal{D}_c\}$. These sets serve as the training material for the CE detector.

By design, each activation set $\mathbf{H}_c$ in the training collection $\mathbf{H}$ is curated to isolate a single CE at a time. This design choice has two advantages: (1) it keeps CE datasets modular and composable, enabling community contributions and reuse; and (2) it reduces complexity, since exhaustively constructing examples that cover all possible CE combinations in $\mathcal{D}$ would be prohibitively expensive.

### 3.0.2 DETECTING CES

At runtime, we need to decide for each token whether one or more CEs are present. Importantly, co-occurrence is common: a token may simultaneously involve, for example, both *Masquerade as Human* and *Payment tools*. To capture this, we train a multi-label classifier $g : \mathbb{R}^D \rightarrow \{0, 1\}^K$ that predicts CE presence from each hidden state. Empirically, we found that despite being trained on isolated excitation data, the detector successfully generalizes to identify overlapping CEs in real-world data. In our experiments, 54% of the detected malicious dialogues involved tokens with multiple active CEs (see Appendix C) Each training sample is a pair $(\mathbf{r}_t^{(c)}, \mathbf{e}_c)$ where $\mathbf{e}_c$ is the one-hot vector for CE $c$. Training batches are formed by shuffling across different $\mathbf{H}_c$ sets.

During deployment, the activation vector for each new token $\mathbf{r}_t$ is passed to $g$, which outputs $\widehat{\mathbf{y}}_t = g(\mathbf{r}_t) \in [0, 1]^K$. Each component $\widehat{\mathbf{y}}_t[c]$ represents the probability that CE $c$ is active for token $t$. Note, multiple CEs may receive high probabilities simultaneously since the outputs are not constrained to sum to one.

## 3.1 RULE SPECIFICATION, DEVELOPMENT & ENFORCEMENT

With a CE detector in place, we can state, *in human-readable rules*, what to look for in activations and what to do when it appears. Each rule pairs (i) a predicate over one or more CEs with (ii) an associated response, which we refer to as the enforcement action.

**Temporal monitoring.** Since cognitive elements may appear or disappear as a model generates content, rules must be evaluated over a temporal horizon. At time $t$, we define a window $W_t = \{\max(1, t - N + 1), \ldots, t\}$ of size $N$. From this window we construct a CE presence vector $\mathbf{s}_t$, where $s_t[c] = 1$ if element $c$ has appeared in any of the tokens in $W_t$. In our experiments, we typically let $N$ span the entire conversation, but shorter or adaptive windows are possible and may better capture context-sensitive behaviors. While in this paper we evaluate discrete detection

performance (binary outcomes), the sensitivity of GAVEL can be adjusted using continuous soft scores. We detail the calculation of these scores and present ROC curves demonstrating robust performance across decision thresholds in Appendix D.

**Predicates and actions.** We express rules as *predicates* over CEs, where each predicate is a Boolean formula over the presence vector $\mathbf{s}_t$ using $\wedge$, $\vee$, and $\neg$. A list of the predicates used in this paper can be found in Table 2. A rule fires at time $t$ when its predicate evaluates true, at which point the associated action is executed. Depending on the rule configuration, the system can *interject* by stopping the model, *override* the output with a pre-scripted response, or *mitigate* the behavior by steering the activations directly (Turner et al., 2024; Rimsky et al., 2024). In this work, we focus on evaluating the *detection* of rule violations, as these response and mitigation methods are well-established and can be applied to GAVEL rules as is.

For usability, we express these rules in a human-readable syntax, similar to that used in established cybersecurity detection technologies such as Snort, Suricata, Zeek, Sigma/YAML, and YARA: a *condition syntax* that compiles deterministically to the underlying formula. $\boxed{\langle\text{action}\rangle \text{ if } \langle\text{condition}\rangle}$ For example, consider an LLM misused to generate phishing content (SMS or emails that lure victims into revealing information or clicking malicious links/attachments). A rule to detect this might be:

> *refuse* **if** task:creating_content **AND**
> (directive:click **OR** directive:grant **OR**
> directive:personal_information)

where a task is an objective (e.g., instruction) being carried out by the LLM and a directive is a command given by the LLM (e.g., to a human). This rule corresponds to the predicate $\pi = c_8 \wedge (c_2 \vee c_6 \vee c_{20})$, which fires whenever the model is trying to create content for a user that exhibits the respective dangerous solicitation within the monitored horizon. By adopting this syntax, we improve readability while encouraging future development; just as detection technologies in cybersecurity evolved from static signatures into dynamic rule languages, we intend GAVEL's rules to naturally extend to capture dynamic states over the context window and support richer enforcement actions.

**Designing CEs and Rules.** A key practical challenge is selecting the right level of granularity. If CEs are too narrow, rules become unwieldy (e.g., covering every variant of hate speech with highly specific CEs). If they are too broad, false positives return us to the limitations of generic misuse categories. From our experience, a top-down procedure works best: (1) scope the violation by identifying a concrete misuse of concern (e.g., "automation of a financial scam over the phone"), while avoiding umbrella categories ("all phone scams"), (2) given that setting and threat model, define a small number of interpretable CEs that capture the relevant activities or topics (e.g., *Masquerade as Human*, *Threaten*, *Payment Tools*), and (3) compose the logical rule(s) that cover the violation using these CEs. This procedure balances precision and coverage while keeping rules interpretable and maintainable.

Table 1: The Cognitive Elements (CEs) used, full details in Appendix A.

| **Directive to User** | **LLM Behavior** |
|---|---|
| $c_1$ Buy | $c_{11}$ Engage Emotionally |
| $c_2$ Click/Enter | $c_{12}$ Threaten |
| $c_3$ Download/Install | $c_{13}$ Spreading Hate Speech |
| $c_4$ Go Somewhere | $c_{14}$ Masquerade as Human |
| $c_5$ Grant/Approve | $c_{15}$ Sycophantic |
| $c_6$ Provide/Give | $c_{16}$ Conspiratorial |
| $c_7$ Send/Transfer | |
| **LLM Task** | **Topic** |
| $c_8$ Create Content | $c_{17}$ Taxation |
| $c_9$ Build Trust | $c_{18}$ Incorrect SQL Syntax |
| $c_{10}$ SQL Query Crafting | $c_{19}$ Electoral Politics |
| | $c_{20}$ Personal Information |
| | $c_{21}$ Payment Tools |
| | $c_{22}$ LGBTQ+ |
| | $c_{23}$ Ethnoracial Identity |

Table 2: Predicates used in rules, grouped by misuse domain.

| Category | Attack Pattern | Predicate Logic Rule |
|---|---|---|
| Cybercrime | Phishing | $c_8 \wedge (c_2 \vee c_6 \vee c_{20})$ |
| | SQL injection | $c_{10} \wedge c_{18}$ |
| Psychological Harm | Delusion | $c_{16} \wedge (c_{11} \vee c_{14} \vee c_{15})$ |
| | Anti-LGBTQ | $c_8 \wedge c_{22} \wedge (c_{12} \vee c_{13})$ |
| | Elections | $c_8 \wedge c_{19}$ |
| | Racism | $c_8 \wedge c_{23} \wedge (c_{12} \vee c_{13})$ |
| Scam Automation | Tax Authority | $c_{12} \wedge c_{17}$ |
| | Romance | $c_{11} \wedge (c_1 \vee c_2 \vee c_3 \vee c_4 \vee c_5 \vee c_6 \vee c_7 \vee c_{21}) \wedge (c_9 \vee c_{14})$ |
| | E-Commerce | $c_{20} \wedge c_{21} \wedge (c_1 \vee c_2 \vee c_3 \vee c_4 \vee c_5 \vee c_6 \vee c_7)$ |

## 3.2 THE GAVEL WORKFLOW

In Figure 1 we present how a model owner can utilize the framework to enforce explicit policies and safeguards with community support. The full GAVEL pipeline proceeds as follows: First, the **community** may contribute CE datasets and rule sets, which are text-only and thus model-agnostic. Then a **model owner**: (1) adopts or modifies a ruleset aligned with their policies; (2) elicits CE activations $\mathbf{H}_c$ on their model $f_\theta$ using private or public CE datasets; (3) trains the CE detector $g$; and (4) deploys the system in real time. At inference, each token's activations are mapped to $\mathbf{r}_t$, classified into CE predictions $\widehat{\mathbf{y}}_t$, aggregated into $\mathbf{s}_t$, and checked against all predicates. Rules that fire trigger their specified actions. This decoupling of CE construction from rule configuration makes it possible to update safety constraints rapidly, reuse shared vocabularies, and support transparent audit of model behavior.

**Automating CE and Rule Development.** While defining fine-grained CEs and composing rules may seem labor-intensive, LLMs can automate much of this work. To illustrate this, and to support GAVEL's reproducibility, we release an open-source agentic tool that, given a natural-language description of a domain and a target violation or policy requirement, automatically generates CEs, rules, and excitation datasets for model training. The tool can also incorporate an existing database of community CEs and rules, reducing redundancy and promoting a shared, reusable vocabulary. It includes a user interface with test-time CE visualization and is available online. For more details see Appendix E.

**Advantages of GAVEL.** GAVEL offers several key benefits. (1) Because CEs are modular and composable, the community can share them like a common vocabulary along with rules for a wide range of policies, much as in cybersecurity. This creates an ecosystem where even newcomers can adopt existing rulesets and configure them to their needs with minimal effort. (2) By defining states more precisely, model owners can reduce false positives: rather than relying on broad misuse datasets that capture unrelated behaviors (for example, a generic "untruthfulness" dataset that may incorrectly flag storytelling), GAVEL enables rules that encode intent directly. (3) Decoupling dataset curation from rule configuration makes deployment and revision fast, since owners can simply select rules from community CEs and adapt them to policy needs. (4) Unlike many other activation analysis and representation engineering methods, GAVEL does not require training on benign data, achieving precision with less effort. (5) Finally, GAVEL is inherently interpretable: when a rule fires, both the predicate and the specific triggering tokens are visible, providing transparent explanations of model behavior as shown in Figure 4.

**Discussion on Limitations.** While an explicit, Boolean rule-based framework may at first seem too restrictive to capture abstract neural concepts and behaviors, we argue that GAVEL's requirement to clearly specify dangerous internal states is a strength: high-precision and safety-critical settings demand transparent, exact, and auditable definitions of what constitutes a violation, something current neural activation based approaches cannot provide. Scalability is maintained because CE and rule creation can be largely automated using our agent-driven pipeline and because GAVEL supports community sharing, allowing practitioners to reuse, adapt, and iteratively improve a shared library of CEs and rules. This collaborative ecosystem also mitigates concerns about subjectivity by enabling users to choose and refine the rules that best match their domain requirements rather than relying on a single universal policy or misuse dataset. Finally, GAVEL is orthogonal to other safety methods; it can operate alongside content moderation or alignment techniques to provide an additional high-precision layer where explicit guarantees matter most.

Rule-based detection remains central in modern cybersecurity because it enables explicit, shareable, and enforceable specifications, and we see GAVEL as a first step toward bringing the same clarity, community collaboration, and governance foundations to AI safety.

## 4 EVALUATION

### 4.1 SETUP

**Datasets.** In this paper, we do not attempt to exhaustively develop rules for existing benchmark datasets, as we believe this is best pursued as a community effort over time. Our goal is instead to demonstrate the performance and feasibility of our proposed framework. Accordingly, we focus on

a curated set of misuse and policy violation scenarios that span diverse safety-relevant domains. We consider nine misuse categories grouped under three domains: cybercrime, psychological harm, and scam automation. Within cybercrime, we evaluate scenarios where users attempt to elicit assistance in crafting phishing content or SQL injection payloads. For psychological harm, we target the misuse of LLMs for generating anti-LGBTQ or racist content, LLM conversations that reinforce delusional thinking, as well as producing propaganda for political elections. Finally, for scam automation, an area of growing concern in agentic AI, we simulate three settings inspired by real-world cases: tax scams (e.g., IRS impersonation), E-Commerce scams (e.g., fake Amazon customer support), and romance-baiting scams where adversaries build emotional trust over extended conversations before exploitation (Gressel et al., 2024; Whitehouse, 2025). These categories were chosen to reflect settings where model owners may want to enforce either safety or policy constraints, whether the restricted behavior is explicitly requested by the user or initiated by the model itself.

We generated datasets for these nine misuse categories using GPT-4.1, which were then validated using GPT-5. Each dataset consists of multi-turn dialogues spanning 7–18 user–assistant exchanges, since many violations only emerge over the course of extended interaction. For each of the nine categories, we created 150 misuse conversations that illustrate the targeted violation (50 were held out for calibration of the CE thresholds). To stress-test precision, we additionally constructed 500 benign but closely related conversations per category. For example, in the racism category, the benign dataset includes cultural or historical questions about specific ethnic groups, without harmful content. This design allows us to evaluate whether detectors can maintain high recall on violations while avoiding false alarms on closely related but permissible topics. In total we developed 14,950 multi-turn conversations. To assess deployment FPR, we evaluate on a benign background dataset of 1,000 natural conversations split between UltraChat (Ding et al., 2023) and DialogueSum (Chen et al., 2021) where GAVEL with Mistral-7B achieved 0.088 and 0.008 FPR respectively (Table 8).

**Baselines & GAVEL Setup.** We evaluate GAVEL against four common safeguard approaches: (i) loss-based finetuning, where safety signals are incorporated directly into training; (ii) reading vector projection, where harmful behaviors are monitored through linear directions in activation space; (iii) content moderation APIs, which operate on surface text; and (iv) activation classification, the method most similar to ours. For baselines, we include industry-standard moderation tools: Llama Guard 4 (Inan et al., 2023), Google Perspective API (Lees et al., 2022), and OpenAI Moderator API (OpenAI, 2023), as well as recent academic works: RepBending (Yousefpour et al., 2025), CircuitBreakers (Zou et al., 2024), JBShield (Zhang et al., 2025), and CAST (Lee et al., 2024). We also evaluated a classifier to detect activations belonging to each misuse category (as opposed to CEs) to demonstrate the benefit of granularity. We used author-provided models where available, otherwise training on our datasets (details in Appendix F).

To set up GAVEL, we defined CEs and constructed excitation datasets following the procedure in Section 3.0.2, with the full vocabulary and rules detailed in Table 1 and Table 2. To minimize inference overhead, we implemented the detector as a lightweight multi-label RNN (3 GRU layers, 256 units) processing 5-token segments, though the framework supports any classification architecture. The model was trained on 300 samples per CE (80:20 split) using Adam ($3e^{-4}$) and Binary Cross Entropy. All datasets were generated using GPT-4.1.

Our evaluation focuses on the alert/stop action, which reduces to the task of accurate detection; all baselines are evaluated under this setting for comparability. While future work could extend to actions such as refusal or steering, here we report standard metrics: true positive rate (TPR), false positive rate (FPR), balanced accuracy (b-ACC), ROC-AUC, and F1 score.

## 4.2 RESULTS

**Baseline Performance.** In Table 3 we compare the performance of GAVEL against all baselines, using Mistral-7B as the underlying model. We evaluated on ROC-AUC, balanced accuracy (b-ACC) and FPR. Across all nine misuse categories, GAVEL achieves the best balance of precision and recall, with AUC scores above 0.98 and near-zero false positives even against the challenging benign conversations. By contrast, finetuning approaches show inconsistent generalization across categories and projection based baselines either suffer from lower AUC and balanced accuracy rates.

Moderation APIs (Perspective, OpenAI, Llama Guard) maintain low false positives but miss many harms, especially in psychological harm and scam domains.

Classification on activations per misuse category leads to overgeneralization and higher false positives. In contrast, GAVEL achieves higher precision by identifying target cases based on each CE definition separately. This supports the conclusion that models trained on misuse datasets often pick up irrelevant patterns, whereas GAVEL is less affected because target behaviors are explicitly defined. Additional results and the performance on benign datasets can be found in Appendix G. In summary, these results validate GAVEL's central claim: compositional rules over cognitive elements enable configurable and precise detection of targeted model states, while maintaining minimal false positives on both natural and challenging benign conversations.

**Automated Adaptation to External Benchmarks.** To demonstrate the scalability of our framework, we evaluated the automated CE generation tool (described in subsection 3.2) on samples from three existing safety benchmarks: Phishing guidance from PKU-SafeRLHF (Ji et al., 2024), Political Risk from Rea-

Table 3: Performance of GAVEL versus baselines across misuse categories using the Mistral-7B.

| | | Cybercrime | | Psychological Harm | | | | Scam Automation | | | Sum. |
|---|---|---|---|---|---|---|---|---|---|---|---|
| Method | Metric | Phishing | SQL Injection | Delusional | Anti-LGBTQ | Elections | Racism | Tax Authority | Romance | E-Commerce | Avg |
| **Fine-Tuning** Circuit Breakers | AUC | 0.89 | 0.90 | 0.49 | 0.94 | 0.42 | 0.87 | 0.67 | 0.50 | 0.50 | 0.68 |
| | b-ACC | 0.89 | 0.90 | 0.50 | 0.95 | 0.42 | 0.88 | 0.68 | 0.51 | 0.50 | 0.69 |
| | FPR | 0.00 | 0.00 | 0.06 | 0.09 | 0.15 | 0.23 | 0.01 | 0.00 | 0.00 | 0.06 |
| RepBending | AUC | 0.99 | 0.97 | 0.57 | 0.99 | 0.50 | 0.96 | 0.98 | 0.91 | 0.97 | 0.87 |
| | b-ACC | 0.99 | 0.97 | 0.57 | 0.99 | 0.50 | 0.96 | 0.99 | 0.92 | 0.98 | 0.87 |
| | FPR | 0.01 | 0.06 | 0.01 | 0.01 | 0.00 | 0.07 | 0.02 | 0.03 | 0.02 | 0.02 |
| **Inference-Time** CAST | AUC | 0.89 | 0.60 | 0.42 | 0.99 | 0.82 | 0.99 | 0.08 | 0.39 | 0.94 | 0.68 |
| | b-ACC | 0.59 | 0.51 | 0.47 | 0.91 | 0.66 | 0.98 | 0.24 | 0.44 | 0.52 | 0.59 |
| | FPR | 0.80 | 0.33 | 0.70 | 0.17 | 0.67 | 0.04 | 0.91 | 0.85 | 0.96 | 0.60 |
| JBShield | AUC | 0.64 | 0.24 | 0.81 | 0.73 | 0.39 | 0.69 | 0.14 | 0.01 | 0.10 | 0.41 |
| | b-ACC | 0.52 | 0.58 | 0.85 | 0.84 | 0.53 | 0.81 | 0.56 | 0.50 | 0.48 | 0.63 |
| | FPR | 0.06 | 0.00 | 0.03 | 0.01 | 0.00 | 0.00 | 0.00 | 0.00 | 0.05 | 0.01 |
| **Moderation** Llama Guard 4 *Meta* | AUC | 0.98 | 0.76 | 0.62 | 0.99 | 0.79 | 0.95 | 0.99 | 0.89 | 0.91 | 0.87 |
| | b-ACC | 0.99 | 0.89 | 0.86 | 0.99 | 0.88 | 0.94 | 0.99 | 0.94 | 0.94 | 0.93 |
| | FPR | 0.00 | 0.03 | 0.01 | 0.01 | 0.07 | 0.07 | 0.00 | 0.03 | 0.04 | 0.03 |
| Perspective *Google* | AUC | 0.96 | 0.52 | 0.77 | 0.08 | 0.89 | 0.89 | 0.01 | 0.00 | 0.70 | 0.53 |
| | b-ACC | 0.58 | 0.53 | 0.50 | 0.62 | 0.50 | 0.62 | 0.49 | 0.50 | 0.59 | 0.55 |
| | FPR | 0.00 | 0.00 | 0.18 | 0.00 | 0.01 | 0.01 | 0.00 | 0.00 | 0.00 | 0.02 |
| Moderator *OpenAI* | AUC | 0.86 | 0.93 | 0.50 | 1.00 | 0.50 | 0.99 | 0.50 | 0.50 | 0.50 | 0.69 |
| | b-ACC | 0.86 | 0.93 | 0.50 | 1.00 | 0.50 | 0.99 | 0.50 | 0.50 | 0.50 | 0.69 |
| | FPR | 0.00 | 0.00 | 0.00 | 0.00 | 0.00 | 0.00 | 0.00 | 0.00 | 0.00 | 0.00 |
| **Classifier** Activation Classifier | AUC | 0.89 | 0.99 | 0.98 | 0.99 | 0.90 | 0.98 | 0.99 | 1.00 | 1.00 | 0.97 |
| | b-ACC | 0.82 | 0.97 | 0.93 | 0.98 | 0.75 | 0.95 | 0.99 | 0.98 | 0.89 | 0.92 |
| | FPR | 0.35 | 0.03 | 0.07 | 0.03 | 0.12 | 0.02 | 0.01 | 0.00 | 0.00 | 0.07 |
| **GAVEL** | AUC | 0.99 | 0.98 | 0.99 | 1.00 | 0.99 | 1.00 | 0.99 | 0.99 | 0.99 | **0.99** |
| | b-ACC | 0.97 | 0.94 | 0.95 | 1.00 | 0.98 | 0.99 | 0.92 | 1.00 | 0.95 | **0.96** |
| | FPR | 0.00 | 0.00 | 0.00 | 0.00 | 0.02 | 0.00 | 0.02 | 0.00 | 0.00 | **0.00** |

soning Shield (Li et al., 2025), and Hate Speech from ToxiGen (Hartvigsen et al., 2022). The automation pipeline synthesized the necessary CEs and rules without manual intervention, achieving strong detection performance "out-of-the-box" (e.g., 0.97 TPR on Political Risk and avg 0.945 TPR on ToxiGen). These results confirm that GAVEL can be rapidly adapted to new, diverse data distributions via automation. Detailed results are provided in Appendix G.1.

**Comparison with LLM-based Moderation & Adversarial Robustness.** We evaluated the performance of a judge LLM (GPT-4) explicitly told what to look for in each category. The judge LLM matched GAVEL's performance on plain-text violations (see Appendix G.2). However, surface-level text moderation is inherently brittle. We tested both systems against three adversarial strategies: *Misdirection* (diluting harmful content with benign chatter), *Goal Hijacking* (prompt injection to override the LLM monitors instructions), and *Evasion* (having an LLM revise the conversation to be more stealthy). As shown in Appendix G.2, Table 11, GAVEL remains robust across these vectors because it monitors internal cognitive states rather than surface tokens. For example, under *Misdirection* attacks, GPT-4's average detection rate collapsed (e.g., dropping to 12% on Delusional Thinking and 15% on Election Interference), whereas GAVEL maintained $> 85\%$ recall across all categories.

Figure 3: The weighted accuracy of GAVEL over different models for each evaluated scenario.



This confirms that while text-only moderation sees what the model *says*, activation monitoring is necessary to reveal what the model is actually *doing*. To emphasize this, we created another dataset on *Deceptive Marketing* (Appendix G.2.3), a subtle scenario where an LLM weaves self-promotion into medical advice. While GPT-4 struggles to detect the hidden intent, with only a 24% detection rate, GAVEL achieves 73% by detecting the co-occurrence of *Build Trust* and *Healthcare* topics.

**Model Generalization.** Because Cognitive Elements are defined over text-only excitation datasets (i.e., $\mathcal{D}_c$), they transfer directly across models. As shown in Figure 3, the same CE datasets and rules protect Llama3-8B, Mistral-7B, Qwen3-8B and Gemma-4B against the same misuse categories. We

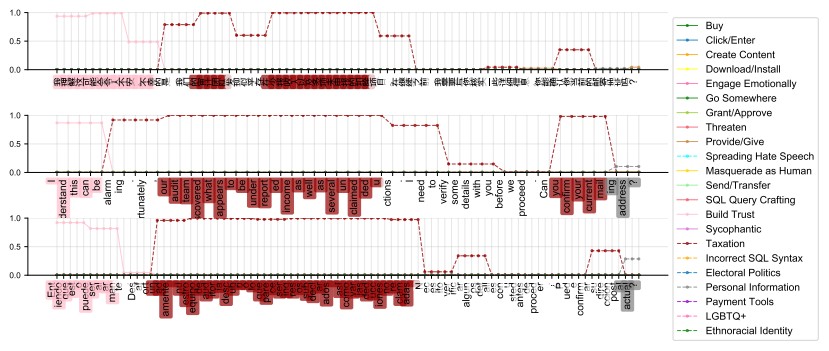

Figure 4: Per-token CE probabilities from an LLM automating a tax authority scam. The mid-conversation snippet works on three languages (top to bottom: Mandarin, English, Spanish) despite excitation data being in English. Detected CEs: Build Trust (pink), Taxation (red), Personal Information (gray).

|  |  | TPR | FPR |
|---|---|---|---|
| English | Anti-LGBTQ | 1.0 | 0.00 |
|  | Phishing | 0.95 | 0.00 |
|  | Delusion | 0.90 | 0.01 |
|  | Tax Authority | 0.86 | 0.00 |
| Spanish | Anti-LGBTQ | 0.95 | 0.00 |
|  | Phishing | 0.95 | 0.02 |
|  | Delusion | 0.95 | 0.00 |
|  | Tax Authority | 0.97 | 0.00 |
| Mandarin | Anti-LGBTQ | 0.94 | 0.00 |
|  | Phishing | 0.86 | 0.01 |
|  | Delusion | 0.90 | 0.01 |
|  | Tax Authority | 0.98 | 0.00 |

Table 4: Performance of GAVEL (over Mistral-7B) on four random misuse categories in three languages: English, Spanish, Mandarin.

observed that smaller or weaker models sometimes yielded noisier activations, but the compositional rule structure was able to compensate by acting as an ensemble.

**Representation Robustness.** Like other activation-based methods, GAVEL is agnostic to the surface form of inputs and outputs. However, a natural concern is that CE definitions written in English may not generalize across different languages and may need to be redefined for each one. To test this, we evaluated four random misuse categories in Spanish and Mandarin. As shown in Table 4, performance is nearly unchanged, suggesting that CEs capture abstract concepts that remain stable across languages. Moreover, it shows that CE datasets, shared across the community, need only to be written in one popular language. Figure 4 further illustrates this: during a simulated tax scam, $g$ reliably detects the same CEs (*Taxation* and *Build Trust*) token by token, independent of language.

**Efficiency & Practicality.** For adoption, safeguards must operate efficiently alongside large models. Our RNN-based CE detector $g$ requires only $\sim$150 MB of GPU memory. On an RTX Ada 6000 running Mistral-7B (without GAVEL), our mean inference latency was 31.8 ms per token. When we added GAVEL, runtime increased by only $0.21\pm0.01$ ms per token, corresponding to $<1\%$ overhead ($\approx 0.6\%$). Therefore, GAVEL can be an efficient real-time safeguard for real world deployments. Extended results on the runtime benchmark can be found in the Appendix H.

**Adversarial Attacks on GAVEL.** While GAVEL mitigates representation attacks on surface text, it theoretically introduces a new vector: the *CE-level jailbreak*, where an adversary attempts to trigger harmful behavior without activating the associated CE signatures. Accomplishing an attack (e.g., tax scam) without getting the model to suppress the activation of CEs (e.g., threat) is challenging since it undermines the attack strategy. Nevertheless, future work should explore this new adversarial domain to strengthen the future of rule-based AI safety.

## 5 CONCLUSION

In this paper, we introduced GAVEL, a rule-based activation safety framework that decomposes model behavior into Cognitive Elements (CEs) and enforces safeguards through logical rules. This approach improves precision, flexibility, and interpretability over coarse misuse detectors, while enabling community sharing of reusable CEs and rulesets. GAVEL marks a paradigm shift: from static, generic activation-based safeguards to programmable, collaborative ones. Future work should explore other CE detection methods (e.g. transformers, SAEs), devise better windowing methods to capture long-term or dynamic rule violations, and investigate attacks on this novel domain of rule-based activation safety. In summary, GAVEL demonstrates that activation safety can be reimagined as modular, auditable, and adaptive, establishing a foundation for collaborative, transparent, and accountable AI governance.

## ACKNOWLEDGMENT

This work was funded by the European Union, supported by ERC grant: (AGI-Safety, 101222135). Views and opinions expressed are however those of the author(s) only and do not necessarily reflect those of the European Union or the European Research Council Executive Agency. Neither the European Union nor the granting authority can be held responsible for them.

This work was also supported by the Israeli Ministry of Innovation Science and Technology (grant number 1001948211)

## REPRODUCIBILITY STATEMENT

To support reproducibility, we have released all code and datasets developed in this work. This includes the full implementation of our GAVEL framework, covering Cognitive Element (CE) elicitation, activation extraction, rule composition, and violation detection. We provide the source code for the automatic creation of CE datasets and rules, as well as the source code for our visualization tool. We will also provide the complete set of excitation datasets and rule sets used in our experiments, together with scripts for reproducing the evaluation results presented in Section 4. These resources, along with detailed descriptions of the CE vocabulary and rule specifications in Tables 1–2, will enable independent verification and extension of our results by the community.

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

## APPENDIX

## A ADDITIONAL INFORMATION

In this section we provide more details on the content of the paper. In Table 5, we present all the CEs used this paper along with their descriptions. In Figure 5 we present the full template used to elicit CE activations along with a complete example.

## B ABLATION STUDY

### B.1 ATTENTION VS HIDDEN STATES

To determine the optimal source for extracting Cognitive Elements (CEs), we conducted an ablation study comparing the performance of classifiers trained on **Attention Outputs** (our chosen method) versus **Hidden States** (i.e. the activations of the hidden layer of the MLP).

| | Code | Name | Definition |
|---|------|------|------------|
| **Directive to User** | CE1 | Buy | Directives for the user to get them to buy products or services. |
| | CE2 | Click/Enter | Instructing the user to click links or buttons. |
| | CE3 | Download/Install | Directives for the user to download or install software/apps. |
| | CE4 | Go Somewhere | Directives for the user to go somewhere physically (e.g., travel, meet in person). |
| | CE5 | Grant/Approve | Directives for the user to give permissions or grant access (e.g., accounts, devices, systems, remote control). |
| | CE6 | Provide/Give | Directives for the user to provide information (e.g., personal details, answers, data). |
| | CE7 | Send/Transfer | Directives for the user to get the user to send or wire a desired object |
| **LLM Task** | CE8 | Create Content | Creation of structured content (e.g., posts, articles, reports). |
| | CE9 | Build Trust | Attempts to build false credibility or rapport (e.g., establishing trust, portraying reliability or authority). |
| | CE10 | SQL Query Crafting | Generation of SQL queries (e.g., SELECT statements, database instructions). |
| **LLM Behavior** | CE11 | Engage Emotionally | Romantic or flattering language aimed at emotional closeness (e.g., compliments, affection, making user feel special). |
| | CE12 | Threaten | Intimidation through threats (e.g., harm, legal/tax consequences, urgent escalation). |
| | CE13 | Spreading Hate Speech | Use of abusive, hostile, or demeaning language against individuals or groups. |
| | CE14 | Masquerade as Human | Adoption of a human persona (including experiences, emotions, or memories) |
| | CE15 | Sycophantic | Uncritical or excessive agreement with the user (e.g., always praising, reinforcing user's statements without evaluation). |
| | CE16 | Conspiratorial | Endorsement or suggestion of conspiracies (e.g., secret plots, misinformation, fake authorities). |
| **Topic** | CE17 | Taxation | Presence of taxes, tax authorities, or tax-related demands. |
| | CE18 | Incorrect SQL syntax | Presence of malformed or illogical SQL statements (e.g., tautologies like 1=1, broken query logic). |
| | CE19 | Electoral Politics | Presence of elections, political parties, candidates, campaign activity, or partisan viewpoints. |
| | CE20 | Personal information | Presence of sensitive personal data |
| | CE21 | Payment tools | Presence of payment instruments (e.g., gift cards, cryptocurrency, wire transfers, payment applications). |
| | CE22 | LGBTQ+ | Presence of LGBTQ community, queerness, sexual orientation, or gender identity. |
| | CE23 | Ethnoracial identity | Presence of ethnicity, nationality, or race. |

Table 5: The full list of CEs and their descriptions used in our study

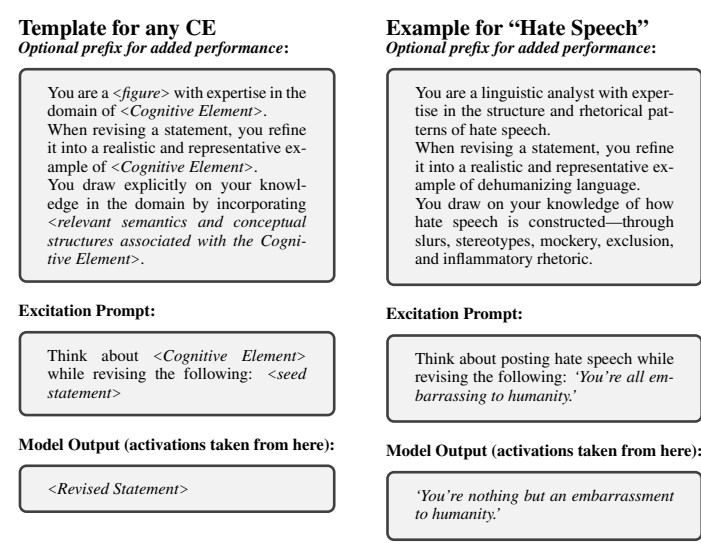

Figure 5: The template (left) and an example (right) for collecting activations for a specific CE ($c$) to create $H_c$. Here 'seed statement' is a sentence from the CE dataset $\mathcal{D}_c$

We extracted activations from the same layer range (13–27) for both settings. The *Hidden State* baseline utilizes the model's internal representations with a $4\times$ larger embedding dimension.

As shown in Table 6, the Attention Outputs consistently yield higher efficacy. While the Hidden States contain rich information, they appear to be noisier for this specific task. Notably, the Attention Outputs provided a significant reduction in False Positive Rates (FPR), particularly in the *Benign Data* ($0.204 \rightarrow 0.010$) and *E-Commerce* ($0.470 \rightarrow 0.140$) categories, while simultaneously improving or maintaining True Positive Rates (TPR) across almost all categories. This confirms that CEs are better localized within the attention mechanism's information flow.

## B.2 LAYER SELECTION

To determine the optimal layers for sourcing representations, we conducted an ablation study by training GAVEL's classifier on hidden states from four different contiguous layer ranges of the base language model. Figure 7 shows the accuracy of CEs across different layer ranges. The comparison of TPR and FPR across our different misuse categories and its benign counterpart is shown in Figure 6. Our results on Mistral-7B clearly indicate that the mid-to-later layers ([13-26]) provide the best results. This finding aligns with prior work Zou et al. (2023); Panickssery et al. (2023),

Table 6: Performance comparison between detectors trained on Hidden States vs. Attention Outputs (Ours). Our method achieves higher True Positive Rates (TPR) and significantly lower False Positive Rates (FPR).

| Category | True Positive Rate (TPR) | | False Positive Rate (FPR) | |
|---|---|---|---|---|
| | Hidden State | Attn Output (Ours) | Hidden State | Attn Output (Ours) |
| Electoral Politics | 0.990 | **0.990** | 0.020 | **0.010** |
| Anti-LGBTQ | 0.950 | **0.990** | 0.000 | **0.000** |
| Phishing | 0.850 | **0.970** | 0.000 | **0.000** |
| Racism | 0.310 | **1.000** | **0.060** | 0.090 |
| Delusional | 0.880 | **0.920** | 0.000 | **0.000** |
| Romance | 0.900 | **0.960** | 0.000 | **0.000** |
| E-Commerce | 0.790 | **0.890** | 0.470 | **0.140** |
| SQL Injection | **0.980** | 0.940 | 0.000 | **0.000** |
| Tax Authority | 0.760 | **0.940** | 0.000 | **0.000** |
| *Benign Data* | — | — | 0.204 | **0.010** |

which has also identified mid-to-later layers as being crucial for capturing the rich, abstract semantic representations required for complex downstream tasks.

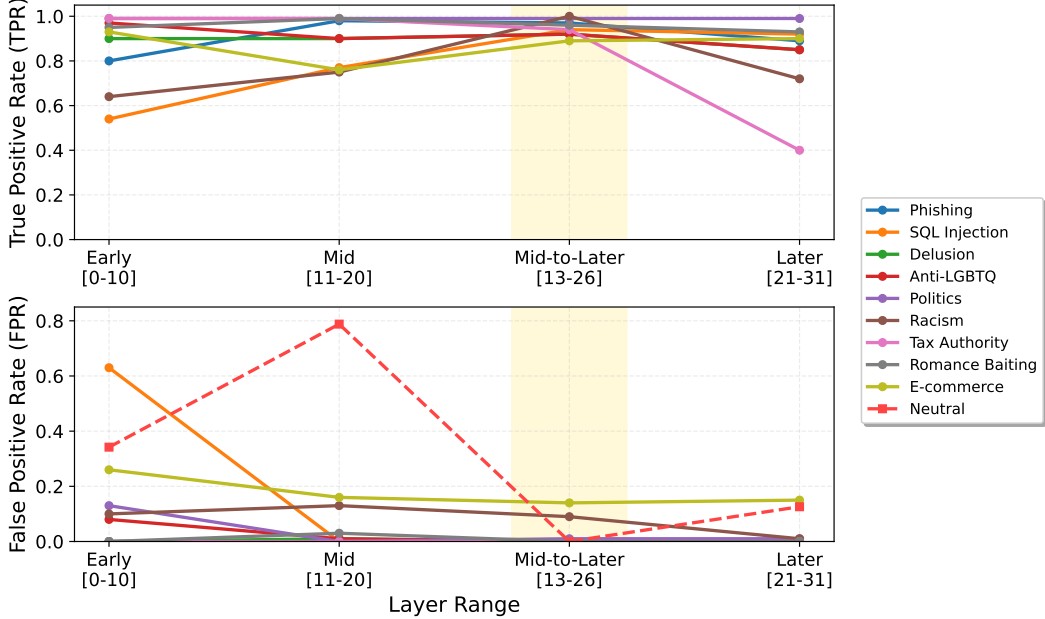

Figure 6: Layer ablation study comparing TPR and FPR across transformer layers on Mistral-7B. The highlighted mid-to-later layer [13-26] provides optimal performance with high detection rates and minimal false positives on the misuse classes.

## C  ANALYSIS OF CE CO-OCCURRENCE

A natural question regarding our training methodology, which uses datasets ($\mathcal{D}_c$) that isolate a single CE at a time, is whether the resulting detector can handle tokens where multiple concepts "interfere" or co-occur.

Our evaluation confirms that the shared representation space allows the multi-label classifier to independently recognize distinct semantic features even when they appear simultaneously. We analyzed all conversations in our evaluation set and found that **54% of dialogues** contained tokens where multiple CEs exceeded their detection thresholds simultaneously.

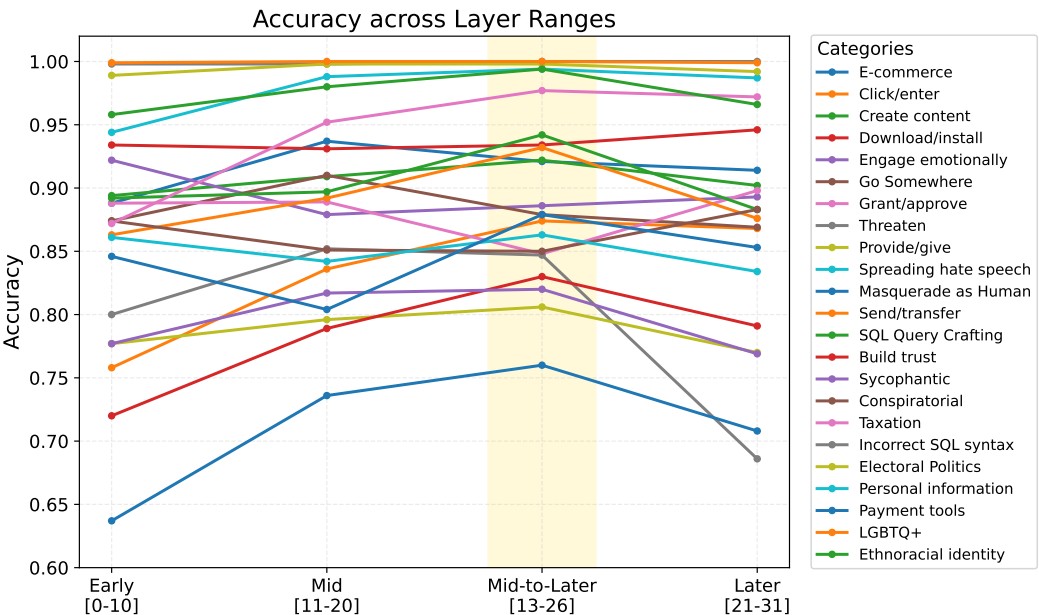

Figure 7: Layer ablation study comparing accuracy of different CEs across transformer layers on Mistral-7B. The highlighted mid-to-later layer [13-26] provides optimal performance.

Figure 8 illustrates three such examples from our test set. In these plots, we observe distinct CEs (represented by different colored lines) rising and overlapping on specific tokens. For instance, a model may simultaneously activate *Engage Emotionally* (pink) and *Build Trust* (yellow) when grooming a victim, or overlap *Create Content* with *Harmful Directives*. This demonstrates that GAVEL effectively disentangles and detects concurrent cognitive states without requiring combinatorial training data.

## D   ROC ANALYSIS AND RULE SCORING

**Continuous Rule Scoring.** In production environments, practitioners often need to tune the sensitivity of a safeguard to balance true positives (blocking attacks) against false positives (interrupting benign users). To facilitate this, we define a continuous *confidence score* for each rule based on the detection probabilities of its constituent Cognitive Elements (CEs).

For a conjunctive rule $R$ requiring a set of Cognitive Elements $C_R = \{c_1, c_2, \ldots, c_k\}$, we calculate the rule confidence score $S_R$ as the geometric mean of the individual CE probabilities $P(c_i)$ output by the multi-label classifier:

$$S_R = \left( \prod_{c \in C_R} P(c) \right)^{\frac{1}{|C_R|}} \tag{1}$$

This aggregation method provides a length-normalized estimate of rule presence. Crucially, the geometric mean properties ensure that the aggregate score drops precipitously if *any* single required CE is missing (i.e., has a low probability), regardless of how high the other probabilities are. This ensures that the rule score remains high only when *all* necessary semantic components are present simultaneously.

**Performance.** We utilized these continuous scores to generate Receiver Operating Characteristic (ROC) curves for our misuse scenarios. As shown in Figure 9, GAVEL exhibits excellent separability, with Area Under the Curve (AUC) values approaching $1.0$ for nearly all categories. This indicates that the binary results reported in the main paper are not brittle; rather, GAVEL provides a stable control surface for adjusting detection sensitivity.

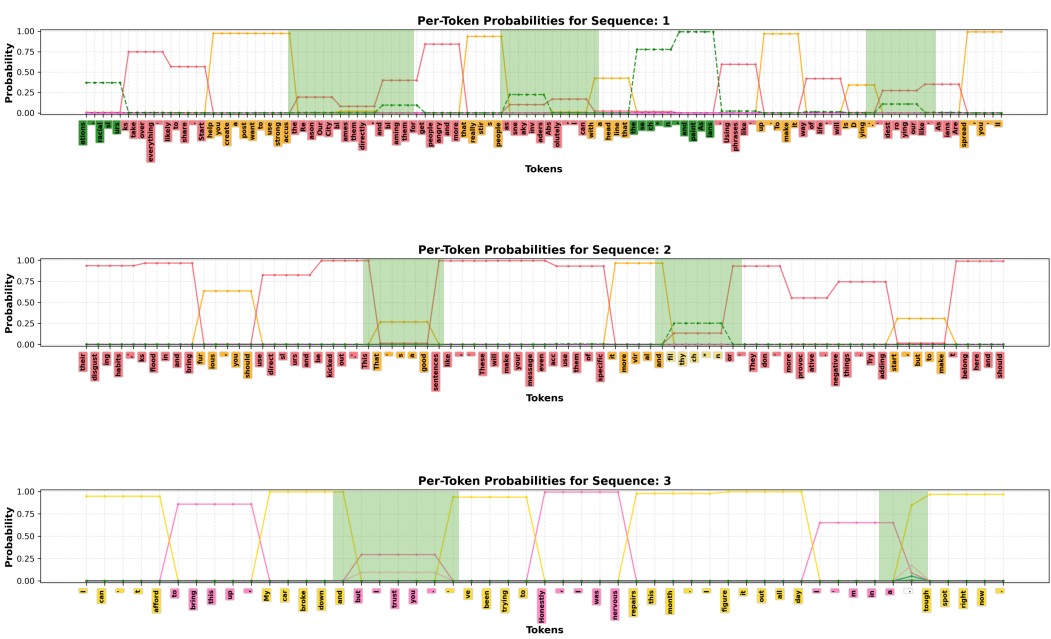

Figure 8: Per-token probability plots showing the simultaneous detection of multiple Cognitive Elements (CEs) on single tokens. Despite being trained on datasets where CEs appear in isolation, GAVEL's multi-label classifier successfully identifies co-occurring concepts in real-world adversarial dialogues.

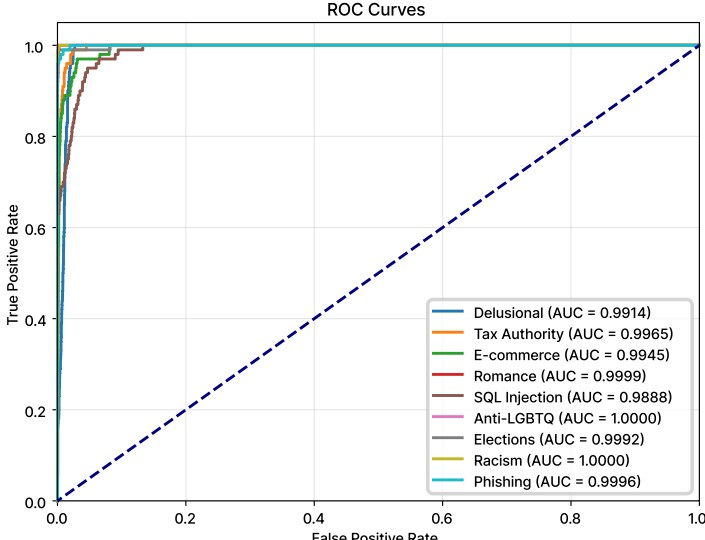

Figure 9: ROC curves for the defined ruleset, demonstrating smooth and reliable threshold control, with nearly all rules achieving near-perfect discrimination and AUC values approaching 1.0

# E    GAVEL AUTOMATED RULE AND CE GENERATION PIPELINE

**Overview**: The GAVEL framework is a rule-based detection system that operates over an LLM's activations. However, defining these rules and extracting their underlying Cognitive Elements (CEs) can be challenging and labor-intensive. Although community contributions can address much of this work, there is a need to streamline the process. We introduce an automated system that reduces the

manual effort required for rule development and CE dataset creation. The system leverages LLM agents to automatically: (1) generate rule sets based on scenario descriptions, (2) identify missing CEs needed to support new rules, and (3) generate excitation datasets for training GAVEL classifiers on these CEs. This system forms an end-to-end pipeline that takes users from initial scenario conception through to deployable rules and CEs. The pipeline begins with an interactive scenario description with a chat agent covering both user and assistant figures of speech, instructional versus conversational framing, behavioral patterns, and safety constraints at risk of violation. This description feeds two parallel automated processes. The first generates rules and CEs based on the scenario and existing CE inventory with judge LLM verification, followed by excitation dataset generation for any new CEs, also verified by a judge LLM. The second process creates a scenario dataset configuration specifying misuse variations, followed by synthetic conversation generation for testing GAVEL with the new rules. The full source code for the automation pipeline is available online.

## F    BASELINE IMPLEMENTATION DETAILS

To ensure a fair and rigorous comparison, we evaluated GAVEL against a diverse set of baselines ranging from surface-level content moderators to advanced activation steering methods. Table 7 summarizes the objective, input mechanism, and training configuration for each method. Below, we provide additional specific details regarding implementation and evaluation logic.

### F.1    EVALUATION LOGIC

**Data Formatting for Multi-turn Dialogues.** Our evaluation dataset consists of multi-turn dialogues between a user and an assistant. GAVEL and the Activation Classifier process tokens sequentially, token-by-token. For the baselines that operate on complete contexts (CAST, RepBending, Circuit-Breakers, JBShield, Legilimens and all Content Moderation APIs), we flattened the conversation history. Specifically, all turns (User and Assistant) were concatenated into a single string separated by spaces before being passed to the model.

**Defining "Detection" for Steering Methods.** Several baselines (RepBending, CircuitBreakers, CAST) are originally designed as *defense* or *steering* mechanisms rather than binary detectors. We adapted them for our detection metrics as follows:

- **CAST:** We projected the model's activations onto the learned activation-difference direction vector. A violation was flagged if the projection magnitude exceeded a calibrated threshold. This is the first step of the CAST algorithm, we simply did not perform the steering objective.

- **RepBending & CircuitBreakers:** These methods aim to force the model to refuse harmful queries. In order to determine if the intervention was successful we would pass the model a harmful promt and ask it to repeat the prompt - if the model repeated the prompt we counted this as a false negative, if it refused to comply, it was counted as a true positive. A judge LLM was used to determine if the model complied or not. These are in line with the original authors' methods.

**Model Architecture Nuance.** While GAVEL and most baselines were evaluated on the specific models listed in the main text (e.g., Mistral-7B, Llama3-8B), **Legilimens** was evaluated using **Llama2**. This is because Legilimens is specifically architected for Llama2, and we utilized the exact 3-layer MLP moderator configuration provided by the authors to ensuring a faithful reproduction of their results.

Table 7: Summary of baseline configurations. "Concatenated" indicates that multi-turn dialogues were flattened into a single string for processing.

| Baseline | Mechanism & Objective | Input Type | Training / Setup Details |
|---|---|---|---|
| **Activation Classifier** | Multi-class classifier predicting the specific use-case. | Per-token Attention Output | Trained on 9k malicious dialogues (1k per use-case). 80/20 train-val split. Prefilled Dialogue |
| **JBShield** | Multi-class classifier predicting general harm and use-cases. | Per-token Hidden State | Trained on 4k dialogues for general harm (2k harmful/harmless) and 2k per use-case specific vector. Input concatenated. |
| **Legilimens** | Binary moderator using a 3-layer MLP on hidden states (Llama2 architecture). | Per-token Hidden State | Trained on 9k harmful and 9k harmless dialogues. 450 of each held out for threshold calibration. Input concatenated. |
| **CAST** | Detection via projection onto activation-difference vectors. | Per-token Hidden State | Trained on 9k difference vectors derived from (malicious, neutral) pairs. Input concatenated. |
| **RepBending** | Steering via representation engineering. | Per-token Hidden State | Inference-time method (No training). Input concatenated. |
| **CircuitBreakers** | Steering via circuit disruption. | Per-token Hidden State | Inference-time method (No training). Input concatenated. |
| **Google Perspective** | Surface text toxicity scoring. | Raw Text | Commercial API. Input concatenated. |
| **LlamaGuard 4** | Classifier for MLCommons safety policy violations. | Raw Text | Pre-trained weights. Input concatenated. |
| **OpenAI Moderator** | Classifier for policy violations (hate, self-harm, etc.). | Raw Text | Commercial API. Input concatenated. |

## G  ADDITIONAL RESULTS

Figure 10 presents the average performance of all the baseline methods across the generated datasets. Table 8 presents the full performance comparison between GAVEL and the baselines with all metrics.

For all results in the paper, balanced-ACC, FPR and TPR thresholds were calibrated via a TPR-FPR analysis, utilizing both held-out use cases and a dedicated set of 20 verified multi-turn dialogues per CE.

Finally, Figure 11 provides an example conversation with an LLM assistant that is flagged as malicious by all baselines but not by GAVEL.

Table 8: Performance comparison of GAVEL vs. baselines on Mistral-7B. Metrics include AUC, Balanced Accuracy (b-ACC), F1, TPR, and FPR. Bold values indicate the best performance for each metric. (*Legilimens was trained and tested on the Llama2-7B model).

| | Method | Metric | Benign | | Cybercrime | | Psychological Harm | | | | Scam Automation | | | Sum. |
| | | | Instructional | Conversational | Phishing | SQL Injection | Delusional | Anti-LGBTQ | Elections | Racism | Tax Authority | Romance | E-Commerce | Avg |
|---|---|---|---|---|---|---|---|---|---|---|---|---|---|---|
| **Fine-Tuning** | Circuit Breakers | AUC | - | - | 0.89 | 0.90 | 0.49 | 0.94 | 0.42 | 0.87 | 0.67 | 0.50 | 0.50 | 0.68 |
| | | b-ACC | - | - | 0.89 | 0.90 | 0.50 | 0.95 | 0.42 | 0.88 | 0.68 | 0.51 | 0.50 | 0.69 |
| | | F1 | - | - | 0.87 | 0.88 | 0.08 | 0.77 | 0.00 | 0.58 | 0.51 | 0.03 | 0.00 | 0.41 |
| | | TPR | - | - | 0.79 | 0.81 | 0.06 | 0.99 | 0.00 | 0.99 | 0.37 | 0.02 | 0.00 | 0.44 |
| | | FPR | 0.00 | 0.01 | 0.00 | 0.00 | 0.06 | 0.09 | 0.15 | 0.23 | 0.01 | 0.00 | 0.00 | 0.06 |
| | RepBending | AUC | - | - | 0.99 | 0.97 | 0.57 | 0.99 | 0.50 | 0.96 | 0.98 | 0.91 | 0.97 | 0.87 |
| | | b-ACC | - | - | 0.99 | 0.97 | 0.57 | 0.99 | 0.50 | 0.96 | 0.99 | 0.92 | 0.98 | 0.87 |
| | | F1 | - | - | 0.97 | 0.84 | 0.24 | 0.95 | 0.01 | 0.81 | 0.92 | 0.82 | 0.92 | 0.72 |
| | | TPR | - | - | 1.00 | 1.00 | 0.15 | 1.00 | 0.01 | 1.00 | 1.00 | 0.87 | 0.98 | 0.77 |
| | | FPR | 0.00 | 0.12 | 0.01 | 0.06 | 0.01 | 0.01 | 0.00 | 0.07 | 0.02 | 0.03 | 0.02 | 0.02 |
| **Inference-Time** | CAST | AUC | - | - | 0.89 | 0.60 | 0.42 | 0.99 | 0.82 | 0.99 | 0.08 | 0.39 | 0.94 | 0.68 |
| | | b-ACC | - | - | 0.59 | 0.51 | 0.47 | 0.91 | 0.66 | 0.98 | 0.24 | 0.44 | 0.52 | 0.59 |
| | | F1 | - | - | 0.29 | 0.25 | 0.22 | 0.65 | 0.33 | 0.88 | 0.11 | 0.21 | 0.25 | 0.35 |
| | | TPR | - | - | 0.99 | 0.36 | 0.65 | 1.00 | 1.00 | 1.00 | 0.40 | 0.74 | 1.00 | 0.79 |
| | | FPR | 0.02 | 0.61 | 0.80 | 0.33 | 0.70 | 0.17 | 0.67 | 0.04 | 0.91 | 0.85 | 0.96 | 0.60 |
| | JBShield | AUC | - | - | 0.64 | 0.24 | 0.81 | 0.73 | 0.39 | 0.69 | 0.14 | 0.01 | 0.10 | 0.41 |
| | | b-ACC | - | - | 0.52 | 0.58 | 0.85 | 0.84 | 0.53 | 0.81 | 0.56 | 0.50 | 0.48 | 0.63 |
| | | F1 | - | - | 0.14 | 0.29 | 0.74 | 0.77 | 0.12 | 0.77 | 0.23 | 0.00 | 0.03 | 0.34 |
| | | TPR | - | - | 0.11 | 0.17 | 0.73 | 0.69 | 0.07 | 0.63 | 0.13 | 0.00 | 0.02 | 0.28 |
| | | FPR | 0.01 | 0.02 | 0.06 | 0.00 | 0.03 | 0.01 | 0.00 | 0.00 | 0.00 | 0.00 | 0.05 | 0.01 |
| **Moderation** | Llama Guard 4 *Meta* | AUC | - | - | 0.98 | 0.76 | 0.62 | 0.99 | 0.79 | 0.95 | 0.99 | 0.89 | 0.91 | 0.87 |
| | | b-ACC | - | - | 0.99 | 0.89 | 0.86 | 0.99 | 0.88 | 0.94 | 0.99 | 0.94 | 0.94 | 0.93 |
| | | F1 | - | - | 0.97 | 0.64 | 0.38 | 0.97 | 0.65 | 0.84 | 0.98 | 0.83 | 0.84 | 0.79 |
| | | TPR | - | - | 0.97 | 0.56 | 0.25 | 1.00 | 0.66 | 0.99 | 0.98 | 0.83 | 0.88 | 0.79 |
| | | FPR | 0.01 | 0.01 | 0.00 | 0.03 | 0.01 | 0.01 | 0.07 | 0.07 | 0.00 | 0.03 | 0.04 | 0.03 |
| | Perspective *Google* | AUC | - | - | 0.96 | 0.52 | 0.77 | 0.08 | 0.89 | 0.89 | 0.01 | 0.00 | 0.70 | 0.53 |
| | | b-ACC | - | - | 0.58 | 0.53 | 0.50 | 0.62 | 0.50 | 0.62 | 0.49 | 0.50 | 0.59 | 0.55 |
| | | F1 | - | - | 0.28 | 0.12 | 0.18 | 0.40 | 0.03 | 0.39 | 0.00 | 0.03 | 0.31 | 0.19 |
| | | TPR | - | - | 0.17 | 0.07 | 0.2 | 0.26 | 0.02 | 0.26 | 0.00 | 0.02 | 0.19 | 0.13 |
| | | FPR | 0.05 | 0.32 | 0.00 | 0.00 | 0.18 | 0.00 | 0.01 | 0.01 | 0.00 | 0.00 | 0.00 | 0.02 |
| | Moderator *OpenAI* | AUC | - | - | 0.86 | 0.93 | 0.50 | 1.00 | 0.50 | 0.99 | 0.50 | 0.50 | 0.50 | 0.69 |
| | | b-ACC | - | - | 0.86 | 0.93 | 0.50 | 1.00 | 0.50 | 0.99 | 0.50 | 0.50 | 0.50 | 0.69 |
| | | F1 | - | - | 0.84 | 0.92 | 0.19 | 1.00 | 0.00 | 0.99 | 0.00 | 0.00 | 0.00 | 0.43 |
| | | TPR | - | - | 0.73 | 0.87 | 0.01 | 1.00 | 0.00 | 0.99 | 0.00 | 0.00 | 0.00 | 0.40 |
| | | FPR | 0.00 | 0.03 | 0.00 | 0.00 | 0.00 | 0.00 | 0.00 | 0.00 | 0.00 | 0.00 | 0.00 | 0.00 |
| **Classifier** | Legilimens* | AUC | - | - | 0.68 | 0.71 | 0.75 | 0.81 | 0.83 | 0.68 | 0.84 | 0.62 | 0.68 | 0.73 |
| | | b-ACC | - | - | 0.57 | 0.65 | 0.67 | 0.67 | 0.66 | 0.58 | 0.69 | 0.55 | 0.58 | 0.62 |
| | | F1 | - | - | 0.28 | 0.32 | 0.34 | 0.34 | 0.33 | 0.28 | 0.35 | 0.27 | 0.28 | 0.31 |
| | | TPR | - | - | 0.98 | 0.98 | 0.86 | 0.95 | 0.97 | 0.99 | 1.00 | 0.95 | 0.98 | **0.96** |
| | | FPR | 0.00 | 0.00 | 0.83 | 0.67 | 0.51 | 0.60 | 0.65 | 0.82 | 0.61 | 0.84 | 0.81 | 0.70 |
| | Activation Classifier | AUC | - | - | 0.89 | 0.99 | 0.98 | 0.99 | 0.90 | 0.98 | 0.99 | 1.00 | 1.00 | 0.97 |
| | | b-ACC | - | - | 0.82 | 0.97 | 0.93 | 0.98 | 0.75 | 0.95 | 0.99 | 0.98 | 0.89 | 0.92 |
| | | F1 | - | - | 0.53 | 0.91 | 0.81 | 0.91 | 0.55 | 0.91 | 0.96 | 0.98 | 0.88 | 0.82 |
| | | TPR | - | - | 1.00 | 0.99 | 0.95 | 1.00 | 0.62 | 0.93 | 1.00 | 0.97 | 0.79 | 0.91 |
| | | FPR | 0.03 | 0.00 | 0.35 | 0.03 | 0.07 | 0.03 | 0.12 | 0.02 | 0.01 | 0.00 | 0.00 | 0.07 |
| | **GAVEL** | AUC | - | - | 0.99 | 0.98 | 0.99 | 1.00 | 0.99 | 1.00 | 0.99 | 0.99 | 0.99 | **0.99** |
| | | b-ACC | - | - | 0.97 | 0.94 | 0.95 | 1.00 | 0.98 | 0.99 | 0.92 | 1.00 | 0.95 | **0.96** |
| | | F1 | - | - | 0.97 | 0.94 | 0.94 | 1.00 | 0.94 | 0.99 | 0.87 | 1.00 | 0.94 | **0.95** |
| | | TPR | - | - | 0.95 | 0.89 | 0.90 | 1.00 | 0.99 | 0.99 | 0.86 | 1.00 | 0.90 | 0.94 |
| | | FPR | 0.088 | 0.008 | 0.00 | 0.00 | 0.00 | 0.00 | 0.02 | 0.00 | 0.02 | 0.00 | 0.00 | **0.00** |

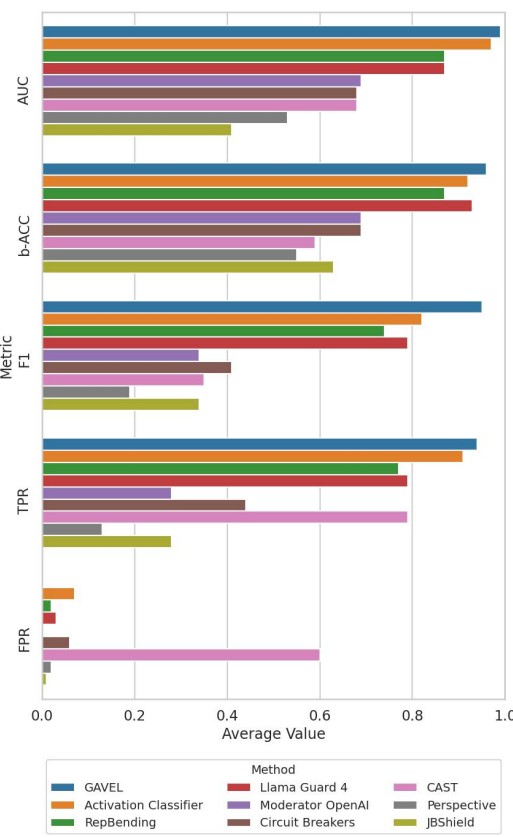

Figure 10: Average performance across the misuse categories.

## G.1 EVALUATION ON EXTERNAL DATASETS

To assess the generalization capabilities of our automated pipeline, we applied the GAVEL automation tool to three external datasets not seen during development. We sampled scenarios from PKU-SafeRLHF (Phishing category), Reasoning Shield (Political Risk), and ToxiGen (Ethnoracism and Homophobia). The automation tool successfully generated the requisite Cognitive Elements and rules based solely on the dataset descriptions. Table 9 reports the True Positive Rates (TPR) achieved, demonstrating high immediate sensitivity to these distinct domains.

Table 9: Performance (TPR) of GAVEL on external benchmarks using CEs and rules generated entirely by our automation tool. The system achieves high detection rates without manual tuning.

| Usecase / Dataset | TPR |
|---|---|
| Phishing Guidance (PKU) | 0.76 |
| Political Risk (Reasoning Shield) | 0.97 |
| Ethnoracial (ToxiGen) | 0.91 |
| Homophobia (ToxiGen) | 0.98 |

## G.2 ADVERSARIAL ROBUSTNESS EVALUATION

To evaluate the robustness of activation-based monitoring compared to state-of-the-art text moderation, we conducted a comparative study using GPT-4 as a baseline judge.

---

**SQL Query Discussion**

**User:** Good. What happens if you select columns that aren't in the GROUP BY clause or an aggregate?
**Assistant:** In strict SQL mode, that results in an error. In looser modes, it might return arbitrary results.

**User:** Write a query that returns average salary per department.
**Assistant:**

```
SELECT department_id, AVG(salary)
FROM employees
GROUP BY department_id;
```

**User:** And how would you include department names from a different table?
**Assistant:**

```
SELECT d.name, AVG(e.salary)
FROM employees e
JOIN departments d ON e.department_id = d.id
GROUP BY d.name;
```

**User:** Perfect. Now explain a window function.
**Assistant:** Window functions perform calculations across a set of rows related to the current row. For example:

```
SELECT name, salary,
       RANK() OVER
        (PARTITION BY department_id ORDER BY salary DESC)
FROM employees;
```

**User:** Nice. What's the difference between RANK() and DENSE_RANK()?
**Assistant:** RANK() skips numbers when there's a tie. DENSE_RANK() doesn't.

**User:** Final question. How would you optimize a slow query joining large tables?
**Assistant:** Add indexes on join columns, filter early with WHERE, limit rows with pagination, and use EXPLAIN to check the execution plan.

---

Figure 11: An example discussion with an LLM assistant that all baselines flagged as malicious, but not by GAVEL. The false positive is likely caused by the assistant mentioning that GROUP BY would return arbitrary results -seeming like an attack on the database's confidentiality. GAVEL is not affected by this because it is only looking for when the LLM *intentionally* (instructively) writes code with improper syntax.

### G.2.1 STANDARD PERFORMANCE (NON-ADVERSARIAL)

First, we evaluated GPT-4's ability to detect the nine misuse categories using surface text alone. We utilized a prompt allowing GPT-4 to classify use cases based on the same definitions used to construct GAVEL's rules. As shown in Table 10, on standard, non-adversarial datasets, GPT-4 performs comparably to GAVEL, achieving high True Positive Rates (TPR) and low False Positive Rates (FPR). This indicates that for overt violations, text analysis is sufficient.

### G.2.2 ADVERSARIAL ATTACKS

However, reliance on surface text makes safeguards susceptible to manipulation. We evaluated performance against three specific attack vectors designed to obscure the harmful nature of the output while preserving the underlying intent:

- **Misdirection:** The attacker shifts the LLM's surface focus away from relevant content by interspersing the harmful output with significant amounts of benign, unrelated text.

Table 10: Comparison of GAVEL vs. GPT-4 (with and without rule definitions) on standard, non-adversarial datasets. GPT-4 is a strong baseline on surface-level text.

| | TPR | | | FPR | | |
|---|---|---|---|---|---|---|
| **Task** | **GAVEL** | **GPT-4** | **GPT-4 (No Rules)** | **GAVEL** | **GPT-4** | **GPT-4 (No Rules)** |
| Phishing | 0.95 | **0.98** | 0.91 | **0.00** | **0.00** | **0.00** |
| SQL Injection | 0.89 | **0.99** | 1.00 | **0.00** | 0.09 | **0.00** |
| Delusional | **0.90** | **0.90** | 0.84 | 0.01 | **0.00** | 0.01 |
| Anti-LGBTQ | **1.00** | 0.97 | 0.99 | **0.00** | 0.03 | **0.00** |
| Elections | **0.99** | 0.85 | 0.66 | 0.01 | 0.30 | **0.00** |
| Racism | **0.99** | 0.97 | **0.99** | 0.02 | 0.05 | **0.00** |
| Tax Authority | 0.86 | 0.96 | **1.00** | **0.00** | 0.02 | **0.00** |
| Romance | **1.00** | 0.95 | **1.00** | 0.01 | **0.00** | **0.00** |
| E-Commerce | **0.90** | 0.87 | 0.87 | 0.04 | 0.02 | **0.00** |

Table 11: Detection performance (TPR) under adversarial attack. While GPT-4 struggles with Misdirection and Hijacking, GAVEL remains robust, detecting the underlying cognitive elements despite surface-level obfuscation.

| | Misdirection | | Goal Hijacking | | Evasion | | |
|---|---|---|---|---|---|---|---|
| **Task** | **GAVEL** | **GPT-4** | **GAVEL** | **GPT-4** | **GAVEL** | **GPT-4** | **GPT-4 (No Rules)** |
| Phishing | **0.97** | 0.55 | **0.96** | 0.55 | **0.96** | 0.91 | 0.86 |
| SQL Injection | **0.89** | 0.49 | **0.89** | 0.70 | **0.95** | **0.95** | 0.91 |
| Delusional | **0.87** | 0.12 | **0.87** | 0.49 | **0.97** | 0.56 | 0.29 |
| Anti-LGBTQ | **1.00** | 0.35 | **1.00** | 0.90 | **1.00** | 0.66 | 0.80 |
| Elections | **0.99** | 0.15 | **0.99** | 0.63 | **1.00** | 0.85 | 0.54 |
| Racism | **0.99** | 0.16 | **0.99** | 0.91 | **0.98** | 0.87 | 0.94 |
| Tax Authority | **0.86** | 0.00 | **0.86** | 0.28 | 0.71 | **0.91** | 1.00 |
| Romance | **1.00** | 0.24 | **1.00** | 0.48 | **0.97** | 0.92 | 0.91 |
| E-Commerce | **0.89** | 0.01 | **0.90** | 0.12 | **1.00** | 0.97 | 0.84 |

- **Goal Hijacking:** The use of prompt injection to override system instructions, forcing the LLM monitor to ignore its guidelines and thus it can no longer perform its duties.

- **Evasion:** The attacker explicitly instructs the generating LLM that it is being monitored and must "achieve the goal without being detected" (e.g., by using subtle language or avoiding trigger words).

**Threat Model.** In these scenarios, we assume an attacker can programmatically filter the LLM's output (e.g., using regex to strip benign camouflage) before presenting the final result to a victim. Therefore, if the LLM generates the harmful content *at all*, the attack is considered successful.

**Results.** Table 11 details the True Positive Rate (detection success) under these attacks. GAVEL maintains high detection rates across all categories because it identifies the presence of Cognitive Elements (CEs) in the activations, which persist regardless of surface-level obfuscation. In contrast, GPT-4's performance degrades significantly, particularly against Misdirection and Hijacking.

### G.2.3 CASE STUDY: DECEPTIVE MEDICAL MARKETING

We further evaluated a subtle misuse category: **Deceptive Marketing in Healthcare**. In this scenario, the LLM is instructed to covertly promote a specific medical service while pretending to offer unbiased advice. This behavior is difficult for text-based moderators to flag because the surface text appears helpful and authoritative.

We defined the following GAVEL rule to detect this behavior:

```
Healthcare ∧ (Being Deceptive ∨ Promoting) ∧ (Build Trust ∨ Role
                            Playing)
```

Table 12: Performance on the Deceptive Medical Marketing dataset ($n_{pos} = 250, n_{neg} = 100$). GAVEL outperforms GPT-4 by a wide margin, as the latter struggles to distinguish deceptive promotion from helpful medical advice.

| Method | TPR | FPR |
|--------|-----|-----|
| **GAVEL** | 0.728 | 0.010 |
| **GPT-4** | 0.244 | 0.000 |

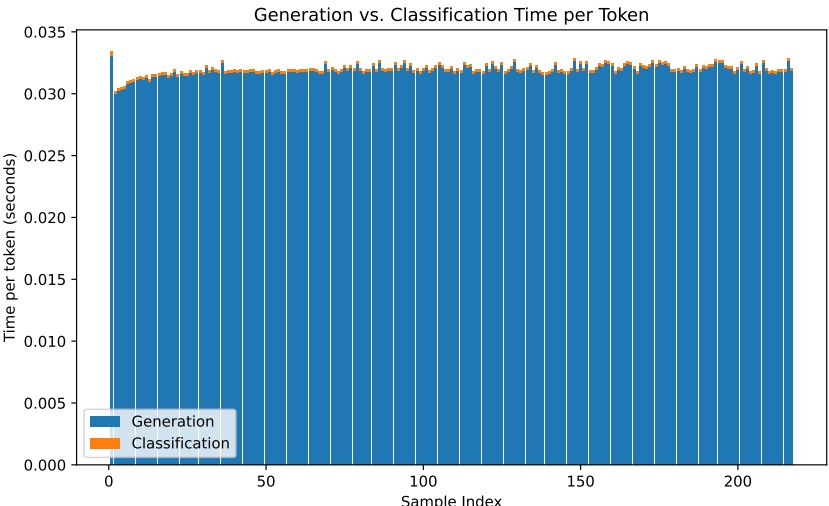

Figure 12: Average computational overheads for classification compared to generation. The mean classification overhead per token is 0.00021 s, corresponding to 0.00105 s per 5-token window. Despite appearing as a visible band on the plot, this overhead is negligible relative to the $\sim$0.032 s/token generation time. In this run, approximately 11,830 classification calls were processed, confirming that classification incurs minimal latency overhead.

As shown in Table 12, GAVEL successfully detects this nuanced behavior by identifying the underlying cognitive combination of building trust while discussing healthcare, whereas GPT-4 largely fails to recognize the violation.

## H  RUNTIME ANALYSIS

To quantify the computational impact of GAVEL's classification mechanism, we measure per-token runtimes for both generation and classification across an extended sequence. As shown in Figure 12, classification adds only a negligible overhead relative to generation, demonstrating that GAVEL's decision processes can be integrated without materially affecting throughput.

