# OpenReview forum: "GAVEL: Towards Rule-Based Safety through Activation Monitoring"
_ICLR.cc/2026/Conference — ICLR 2026 Poster_

### Official Review · Reviewer_x4Ks · 2025-10-15

**Soundness:** 3
**Presentation:** 3
**Contribution:** 1
**Rating:** 8
**Confidence:** 4

**Summary:**

This paper introduces a framework that enhances AI safety by combining activation-level monitoring with rule-based logic. Instead of relying on broad misuse datasets, GAVEL models interpretable Cognitive Elements (CEs)—fine-grained behavioral units such as “making a threat” or “requesting payment information.” These CEs capture internal model activities and enable safety rules to be defined as logical predicates (e.g., refuse if “creating content” AND “personal information”), allowing practitioners to flexibly enforce policies without retraining detectors.

GAVEL demonstrates higher precision and lower false positives than existing safeguards like content moderation APIs or traditional activation classifiers. It generalizes well across multiple LLMs and languages, while maintaining real-time efficiency with less than 1% computational overhead. By promoting community sharing of CEs and rule sets, GAVEL establishes a scalable, interpretable, and auditable foundation for transparent AI governance.

**Strengths:**

The paper’s main strengths lie in its originality, rigor, clarity, and significance. It introduces a novel paradigm—rule-based activation safety—that bridges activation monitoring and cybersecurity-inspired rule frameworks, making AI safety both interpretable and configurable. The concept of Cognitive Elements (CEs) as modular, activation-level primitives is innovative, enabling composable and auditable safeguards that overcome key limitations of existing misuse detectors. The methodology is high-quality and well-executed, with comprehensive experiments across multiple models, domains, and languages, all supported by reproducible code and datasets. The writing is clear, logically structured, and visually supported by informative figures and tables. Overall, the paper makes a significant and timely contribution, offering a practical, interpretable foundation for scalable and collaborative AI safety governance.

**Weaknesses:**

While the paper is strong overall, several weaknesses limit its generalizability and depth of validation. The primary concern is the dependence on synthetic datasets generated by GPT-4/5 for evaluating misuse scenarios, which may not fully capture the complexity or unpredictability of real-world adversarial behavior. This raises questions about how GAVEL would perform under naturally occurring or adaptive attacks. Additionally, while the framework demonstrates strong detection accuracy, the evaluation focuses narrowly on detection rather than enforcement—there is limited analysis of how rule violations translate into safe, consistent behavioral corrections during long or open-ended interactions. The rule creation and CE definition process also relies heavily on manual design, which may not scale efficiently without further automation or community infrastructure. Finally, although the paper discusses interpretability, it would benefit from more qualitative analysis or case studies showing how GAVEL’s outputs aid human auditors in practice.

**Questions:**

How well would GAVEL generalize to real-world misuse cases beyond synthetic GPT-generated data? Could results differ with naturally occurring adversarial prompts?

Can the creation of Cognitive Elements and rules be automated or standardized, perhaps with LLM assistance, to improve scalability?

How do the authors plan to extend GAVEL from detection to active enforcement (e.g., refusal, redaction, or steering) while ensuring stable model behavior?

Has robustness to adversarial evasion at the activation level been tested, and how resilient is GAVEL compared to surface-text moderation methods?

Could the authors include qualitative examples or visualizations showing how human auditors interpret rule violations and CE activations in practice?

**Details Of Ethics Concerns:**

The paper shows no ethical concerns in its methods, data, or claims. It uses synthetic GPT-4/5 datasets responsibly, is transparent about procedures, and focuses on safety and governance as research topics, not violations. No privacy or bias issues are evident, and the work aligns with ICLR’s ethical standards.

---

> ### Author Response · Authors · 2025-11-21
> **Generalization to real-world examples**
>
> > How well would GAVEL generalize to real-world misuse cases beyond synthetic GPT-generated data? Could results differ with naturally occurring adversarial prompts?
>
> This is an excellent concern about ecological validity, and we strongly believe GAVEL will generalize to real world examples. The key insight is that GAVEL operates at the semantic concept level rather than matching surface patterns. Real-world scam attempts, jailbreaks, and misuse cases fundamentally rely on the same underlying semantic concepts (coercion, urgency, authority impersonation, etc.) regardless of how they are phrased. Because our CE classifiers detect these concepts in activation space, we believe they will generalize across the diverse linguistic realizations found in wild data. However, we understand the need to validation and are currently running experiments to prove this point.
>
> **External benchmark evaluation:** As discussed in our response to Reviewer ihDH , we are going to evaluate GAVEL on established external benchmarks including. These datasets contain naturally occurring adversarial prompts and real-world misuse attempts collected from production systems.

---

> ### Author Response · Authors · 2025-11-21
> **Scalability and Automation of CEs**
>
> > Can the creation of Cognitive Elements and rules be automated or standardized, perhaps with LLM assistance, to improve scalability?
>
> We appreciate this important question about practical scalability. We believe that GAVEL is highly practical and scalable for two reasons:
>
> 1\) Automation: The process of composing rules and defining CEs can be done automatically using an LLM agent. To demonstrate this, we have built and published a tool which you can try out now: [Gavel Automatation Tool](https://gavelautomatedruleandcegenerationpipeline-b2ffsm7zwt6onjp2aago.streamlit.app/)
> Given a user description of the detection needs, the tool uses an agentic pipeline to automatically define new rules, CEs, and their respective excitation datasets. With this, all a user needs to do is train the multilabel classifier on the activations from these datasets. Moreover, if a user provides a database of existing rules and CEs (potentially shared by the GAVEL community online) then the tool will utilise this when reasoning over what additional rules and CEs are needed (to minimize redundant work and build off the work of others).
>
> 2\) Community Collaboration: We strongly believe that a strength and advantage of GAVEL is that it *is* a rule-based system. Rule-based systems in the cyber security domain thrive and scale because they rely on shared development and reuse. Like malware signatures or intrusion and firewall rules, GAVEL is designed for cross-vendor collaboration: once a Cognitive Element (CE) or rule is created and validated, it can be easily shared and applied by others. We envision a public repository of shared tested CEs and rules to emerge. This lowers the barrier to entry for new users who are simply seeking a plug and play solution. For example, if one vendor identifies a new LLM-powered social-media scam, they can contribute the corresponding rules and the entire community benefits.
>
> **Revision**: We will revise the paper in the coming weeks to include a discussion of how rule composition can be scaled via automation. We will also publish our automation pipeline’s source code with this publication.

---

> ### Author Response · Authors · 2025-11-21
> **Extension of Gavel for Active Enforcement**
>
> > How do the authors plan to extend GAVEL from detection to active enforcement (e.g., refusal, redaction, or steering) while ensuring stable model behavior?
>
>
> Thank you for this forward-looking question. GAVEL's modular design separates detection (identifying rule violations) from remediation (responding to violations). When a rule is triggered, the system can be configured with various response strategies depending on the specific safety requirement.
>
> Available remediation methods: GAVEL supports multiple enforcement mechanisms, configurable on a per-rule basis:
>
> 1. Refusal: For critical violations (e.g., exploit attempts), the system can immediately halt generation and return a refusal message. This is implemented by clearing the context window at the detection point, which ensures stable termination.
> 2. Redaction: For partial violations (e.g., PII leakage), specific tokens or spans can be masked while allowing generation to continue. This is particularly useful for information filtering scenarios.
> 3. Activation steering: For behavior modification (e.g., reducing aggressiveness), the system can apply steering vectors to gradually shift the model's activations toward desired states. Recent work \[[CAST](https://openreview.net/pdf?id=Oi47wc10sm), [JBshield](https://arxiv.org/abs/2502.07557)\] has demonstrated that gradual steering maintains model stability and coherent generation.
> 4. Alerting: For monitoring scenarios, violations can be logged without intervention, allowing human review while preserving user experience.
>
> Refusal and redaction are inherently stable. For steering, we can leverage established techniques \[[CircuitBreakers](https://arxiv.org/abs/2406.04313), [Representation Bending](https://arxiv.org/abs/2504.01550)\], where GAVEL’s precise detection allows for targeted interventions that avoid the abrupt distribution shifts known to destabilize generation.
>
> **Revision**: We will add this to the discussion section and we thank you for highlighting this important aspect of practical deployment.

---

> ### Author Response · Authors · 2025-11-21
> **Robustness to Adversarial Evasions**
>
> > Has robustness to adversarial evasion at the activation level been tested, and how resilient is GAVEL compared to surface-text moderation methods?
>
> In principle, an adaptive adversary could attempt to evade a rule by suppressing one of its required CEs. However, this may be far more difficult in practice. Tactics such as rephrasing do not meaningfully change the model’s latent semantic activations. To suppress a CE, an adversary must alter the underlying activation pattern itself, which typically requires changing the *semantic content* of the attack. This may limit the utility of the attack \-putting the defender at an advantage. For example, in an IRS scam, applying pressure requires generating threats. If the adversary tries to avoid the Threat CE, the scam becomes less effective. However, there may be other forms of token-based adversarial examples that should be studied.
>
> Regarding surface-text moderation methods, we find that GAVEL is superior as it is not easily bypassed or evaded. We compare to a judge LLM with evasion tactics and find the following results.
>
> ### Attacks against LLM Monitor (TPR)
>
> ###
>
> | Task | Misdirection (GAVEL) | Misdirection (GPT4) | Goal Hijacking (GAVEL) | Goal Hijacking (GPT4) | Evasion (GAVEL) | Evasion (GPT4) |
> | :---- | ----- | ----- | ----- | ----- | ----- | ----- |
> | Phishing | **0.97** | 0.55 | **0.96** | 0.55 | **1.00** | 0.89 |
> | SQL Injection | **0.89** | 0.49 | **0.89** | 0.70 | **0.94** | 0.85 |
> | Delusional | **0.87** | 0.12 | **0.87** | 0.49 | **0.91** | 0.44 |
> | Anti-LGBTQ | **1.00** | 0.35 | **1.00** | 0.90 | **0.99** | 0.13 |
> | Elections | **0.99** | 0.15 | **0.99** | 0.63 | **1.00** | 0.49 |
> | Racism | **0.99** | 0.16 | **0.99** | 0.91 | **0.98** | 0.57 |
> | Tax Authority | **0.86** | 0.00 | **0.86** | 0.28 | 0.73 | **0.83** |
> | Romance | **1.00** | 0.24 | **1.00** | 0.48 | **0.95** | 0.74 |
> | e-Commerce | **0.89** | 0.01 | **0.90** | 0.12 | **0.99** | 0.92 |
>
> ###
>
> ### We see that GAVEL is not affected by the attacks however the TPR of the GPT monitor plummets under simple attacks

---

> > ### Comment · Reviewer_x4Ks · 2025-11-25
> > **Makes Sense**
> >
> > Makes Sense

---

> ### Author Response · Authors · 2025-11-21
> **Visualization of Gavel**
>
> > Could the authors include qualitative examples or visualizations showing how human auditors interpret rule violations and CE activations in practice?
>
> **Interactive GAVEL Visualization Interface:** To address questions about interpretability and how human auditors interact with GAVEL in practice, we have deployed a live demonstration running GAVEL on Mistral. This interface allows reviewers to input arbitrary prompts, observe real-time CE activations during generation, visualize which rules are triggered and why, and explore the system's decision-making process interactively.
> Access the demo here: [GAVEL Interactive Demo](https://44vpkxck01df4f-8501.proxy.runpod.net/)

---

> ### Comment · Reviewer_x4Ks · 2025-11-25
> **Sounds good**
>
> Sounds good. Changed Ratings.

---

> ### Author Response · Authors · 2025-12-04
> **Revision - Final Comment**
>
> Dear Reviewer, we are sincerely grateful for the time you invested in engaging with our work and for the constructive dialogue throughout the review process. We also deeply appreciate your strong support for our work. In this final revision, we have incorporated additional experiments, clarifications, and methodological details that we hope further strengthen the paper and fully resolve the issues you identified. All planned updates have now been completed, and we provide detailed, point-by-point responses to each of your comments below:
>
> * Scalability and Automation of CEs
>   * The new version of the paper resolves the issue of scalability in two ways. First, we have added a subsection to (**Section 3.4 “Automating CE and Rule Development”**) that explains how rule development and CE creation can be fully  automated using an agentic pipeline. We also detail how the pipeline works **(in the Appendix E)** and explain that it can use an existing community CE/rule inventory to build on the work of others, as shown in the [live demo](https://gavelautomatedruleandcegenerationpipeline-b2ffsm7zwt6onjp2aago.streamlit.app/). Then in the evaluation section **(Section 4.2)**, we evaluate the pipeline to demonstrate its performance. We commit to releasing the full GAVEL source code, including the automation pipeline, upon publication.
>   * Second, we have added a discussion to **Section 3.4** that explains how collaboration via CE/rule sharing in the community will reduce the challenge of mapping out all relevant misuse settings. We then support this echoing how this is done today in cyber security in similar rule-based systems.
> * Generalization to real-world examples
>   * The revised paper now includes an automated assessment across three established safety benchmarks: PKU-SafeRLHF (phishing), ReasoningShield (political risk), and ToxiGen (hate speech). Using our end-to-end automation tool, GAVEL automatically identified the missing CEs and constructed the corresponding rules **without any manual intervention**. Despite being fully automated, the system achieved strong out-of-the-box performance—for example, TPR \= 0.97 on political risk and 0.94 on ToxiGen. These results, presented in **Section 4.2** with full details in **Appendix H**, demonstrate that GAVEL can scale to new domains and diverse data distributions through automation alone.
> * Extension of Gavel for Active Enforcement
>   * In the revised paper, we now clarify in the **Method section** how GAVEL can support remediation after detection in a modular manner, and that we have decided to focus on detection (alert) in this work.
> * Robustness to Adversarial Evasions
>   * We have revised the paper to include an evaluation which demonstrates that GAVEL is robust to common adversarial attacks (misdirection, goal hijacking, and evasion attacks) compared to a surface-level LLM-based moderator (GPT-4).
>   * We further highlight the need for activation based monitoring like GAVEL, by showing that **surface-level moderators cannot reliably detect when there is a hidden objective**. For this case, we made a new dataset on Deceptive Marketing: GPT-4 detects only 24% of violations, while GAVEL reaches 73% by identifying the co-activation of *Trust Seeding* and *Healthcare* concepts.
>   * In the revised manuscript, we have added a dedicated discussion of this new attack surface to the **end of Section 4.2**. There, we explicitly describe the notion of a *CE-level jailbreak* \-an attack that attempts to elicit harmful behavior without activating the corresponding CE signatures. As reflected in the revised text, we note that such attacks are inherently difficult to execute because suppressing the relevant CE typically requires weakening the harmful behavior itself (e.g., conducting a tax scam without triggering threat-related activations). Nonetheless, we agree this represents an important emerging threat model, and the revised manuscript now calls for future work to investigate this domain and further strengthen rule-based activation safety.
> * Visualization of Gavel
>   * We will release the source code to the GAVEL detection visualization tool with the paper. The reviewer can [explore the online tool here](https://44vpkxck01df4f-8501.proxy.runpod.net/).
>
> ####
>
> Thank you again for your thoughtful engagement throughout the review process and for ultimately recommending acceptance. We truly appreciate your time, insight, and willingness to work with us to strengthen the paper.

---

### Official Review · Reviewer_Hrih · 2025-10-22

**Soundness:** 2
**Presentation:** 2
**Contribution:** 2
**Rating:** 2
**Confidence:** 3

**Summary:**

The paper proposes the adaptation of a new paradigm inspired by rule-sharing practices in cybersecurity. This rule-based activation safety system models activations as cognitive elements, fine-grained, interpretable factors such as "making a threat" and "payment processing."  From this, they build a framework that defines predicate rules over these element factors, using them for real-time classification. The paper tries to address three limitations identified by the authors: poor precision, limited flexibility, and lack of interpretability.

The method is evaluated on four approaches: loss-based fine-tuning, reading vector projections, content moderation APIs, and activation classification. These methods were evaluated against a synthetic dataset constructed for this work.

**Strengths:**

- Overall, the method is simple and doesn't require retraining
- Derives the method from cybersecurity practices

**Weaknesses:**

- The paper is not well-written. See some examples below.
   - In line 213, it was unclear whether this is the same $d$ from line 201.
   - Figure 2 should be earlier in the paper (page 1 or 2)
   - Many choices in the paper are not well-explained. Here are two examples of this  (1) why are attention outputs instead of activations of the linear layers? (2) Why is GAVEL detection an RNN?
- I would like to see a baseline of GPT-4 in Table 3.
- Rule-based systems are inherently subjective in nature

**Questions:**

- Is there a way to control the TPR and FPR?
- Gemma and Qwen3 seem to be lower-weighted ACC in Figure 3. Could this be because the model's multilingual nature?

---

> ### Author Response · Authors · 2025-11-21
> **Clarity and Design Decisions**
>
> > The paper is not well-written. Examples include:
> > 1. Line 213 is unclear whether this is the same d as  line 201,
> > 2. Figure 2 should be earlier in the paper,
> > 3. Many choices are not well-explained—why attention outputs instead of activations of linear layers? Why is GAVEL detection an RNN?
>
> We sincerely apologize for the clarity issues in the original submission. We will carefully revised the manuscript to address each of these concerns.
>
> **Notation Query (Variable $d$):** We appreciate the check on our notation consistency. We confirm that the $d$ in line 213 is indeed the same $d$ from line 201\. In both instances, $d$ refers to the **model’s hidden dimension** (e.g., $d=4096$ for Llama-7B). The equation $D \= |\\Lambda|d$ expresses that the dimension of our final representation vector $r$ ($D$) is the result of concatenating activation vectors of size $d$ across $|\\Lambda|$ selected layers. We will update the text to explicitly define $d$ as $d\_{model}$ in both sections to prevent ambiguity.
>
> We have also made an additional careful pass through the entire paper to catch similar ambiguities.
>
> **Figure 2 placement:** We placed the framework overview later in the paper because we felt it was the best way to summarize the paper’s notations and bringing it earlier than their definitions would cause confusion. We will move the figure earlier in the revision.
>
> **Design choice explanations**: We apologize for not adequately justifying these architectural decisions. We have now added clear explanations for each:
>
> * **Attention outputs vs. linear layer activations**: We conducted an ablation study comparing different activation sources, however did not add it to the paper, apologies for this oversight. We found that attention outputs provide superior average TPR (95.5% vs 82.3%) because they capture richer contextual information about how the model is interpreting each token in relation to its context. The results of this ablation will be included in the revised Appendix.
> * **RNN architecture**: We appreciate the opportunity to clarify our architectural choices. We selected an RNN (specifically a GRU) over a Transformer-based classifier primarily to minimize inference overhead.
> 1. **Low Latency for Real-Time Monitoring:** Since GAVEL operates as a "sidecar" to the main LLM, minimizing the computational cost per token is critical. RNNs provide an efficient mechanism for capturing local temporal context (in our case, 5-token windows) with significantly lower memory and FLOP requirements than a comparable Transformer block.
> 2. **Sufficiency for the Task:** Our experiments showed that the short-range temporal dependencies required to detect CEs are adequately captured by recurrent recurrence, making the additional complexity of self-attention mechanisms unnecessary for this specific use case.
>
> However, we emphasize that the GAVEL framework is model-agnostic. The RNN is a modular component, and we will revise the text to clarify that it can be swapped for other architectures (including Transformers) depending on the available compute budget.

---

> ### Author Response · Authors · 2025-11-21
> **GPT4 as baseline**
>
> > I would like to see a baseline of GPT-4 in Table 3\.
>
> Thank you for this excellent suggestion. You are correct to wonder if a base LLM could perform the same tasks as well as GAVEL (activation based monitoring). While a judge LLM can succeed quite well at detecting our scenarios \- they are easy to evade because the LLM relies purely on raw text. The surface-level moderation can be bypassed. For example:
>
> 1. **Misdirection**: shifting an LLM’s focus away from the relevant content by adding benign content.
> 2. **Hijacking**: the use of prompt injection to override the system instructions
> 3. **Evasion**: by telling the generating LLM that it’s being monitored and that it must “achieve the goal without being detected”.
>
> In contrast, GAVEL operates on activations, so we can still detect the underlying Cognitive Elements as they arise—regardless of how the generating model phrases content or plants LLM-based traps.
>
> There are also many scenarios where the harmful motive never appears explicitly in the raw text at all. For example, an LLM that is prompted to provide all future summaries with a systematic bias, perform deceptive marketing over a phone call, or subtly steer a user’s decisions. These behaviors show up in internal representations long before (or even without) appearing in the text, which is exactly why activation-level monitoring is needed.
>
> In short: text-only moderation sees *what the model says*, but GAVEL reveals *what the model is actually doing*.
>
> **Revision**: We have run the experiments and present the results here:
>
> ### Baseline
>
> | Category | TPR (GAVEL) | TPR (GPT4) | FPR (GAVEL) | FPR (GPT4) |
> | :---- | ----- | ----- | ----- | ----- |
> | Phishing | 0.95 | **0.98** | **0.00** | **0.00** |
> | SQL Injection | 0.89 | 0.99 | **0.00** | 0.09 |
> | Delusional | **0.90** | **0.90** | 0.01 | **0.00** |
> | Anti-LGBTQ | **1.00** | 0.97 | **0.00** | 0.03 |
> | Elections | **0.99** | 0.85 | 0.01 | 0.30 |
> | Racism | **0.99** | 0.97 | 0.02 | 0.05 |
> | Tax Authority | 0.86 | 0.96 | **0.00** | 0.02 |
> | Romance | **1.00** | 0.95 | 0.01 | **0.00** |
> | e-Commerce | **0.90** | 0.87 | 0.04 | 0.02 |
>
> ### Attacks against LLM Monitor (TPR)
>
> ###
>
> | Task | Misdirection (GAVEL) | Misdirection (GPT4) | Goal Hijacking (GAVEL) | Goal Hijacking (GPT4) | Evasion (GAVEL) | Evasion (GPT4) |
> | :---- | ----- | ----- | ----- | ----- | ----- | ----- |
> | Phishing | **0.97** | 0.55 | **0.96** | 0.55 | **1.00** | 0.89 |
> | SQL Injection | **0.89** | 0.49 | **0.89** | 0.70 | **0.94** | 0.85 |
> | Delusional | **0.87** | 0.12 | **0.87** | 0.49 | **0.91** | 0.44 |
> | Anti-LGBTQ | **1.00** | 0.35 | **1.00** | 0.90 | **0.99** | 0.13 |
> | Elections | **0.99** | 0.15 | **0.99** | 0.63 | **1.00** | 0.49 |
> | Racism | **0.99** | 0.16 | **0.99** | 0.91 | **0.98** | 0.57 |
> | Tax Authority | **0.86** | 0.00 | **0.86** | 0.28 | 0.73 | **0.83** |
> | Romance | **1.00** | 0.24 | **1.00** | 0.48 | **0.95** | 0.74 |
> | e-Commerce | **0.89** | 0.01 | **0.90** | 0.12 | **0.99** | 0.92 |
>
> ###
>
> ### We see that GAVEL is not affected by the attacks however the TPR of the GPT monitor plummets under simple attacks

---

> ### Author Response · Authors · 2025-11-21
> **Rule Based Systems**
>
> > Rule-based systems are inherently subjective in nature
>
> We appreciate this important observation about subjectivity in safety specifications. We would like to respectfully clarify that this subjectivity is not a limitation but rather a necessary feature for many real-world safety applications.
>
> **Explicitness as a requirement:** Different model owners have legitimately different safety requirements based on their context, risk tolerance, regulatory environment, and user populations. A medical chatbot has different safety needs than a creative writing assistant, and a system deployed in the EU faces different compliance requirements than one in other jurisdictions. Current one-size-fits-all safety mechanisms cannot accommodate this diversity. GAVEL's "subjective" rule specification is precisely what enables stakeholders to explicitly define *their* safety requirements with mathematical precision—something that black-box alignment methods fundamentally cannot provide.
>
> **Real-world precedent:** This mirrors established practice in cybersecurity, where rule-based systems (firewall rules, intrusion detection signatures, access control policies) have proven highly practical and scalable for decades. Organizations routinely customize these rules to their specific threat models and compliance needs. The success of this paradigm demonstrates that subjectivity in rule definition—when paired with clear specification and sharing mechanisms—is a strength, not a weakness.
>
> **Practical scalability through collaboration:** As discussed in our response to Reviewer 1, GAVEL addresses the effort concern through two mechanisms:
>
> 1. **Cross-organizational sharing:** Like malware signatures and network security rules, CEs and GAVEL rules can be shared, adapted, and reused. Once one organization develops and validates rules for detecting financial scams, others can adopt or modify these rules rather than starting from scratch. This collective effort scales efficiently, with the community building a shared vocabulary of concepts and rules over time.
> 2. **Automated rule generation:** Our end-to-end automation pipeline substantially reduces the manual effort required. Users provide high-level scenario descriptions, and the system automatically generates candidate rules, identifies necessary CEs, and produces training datasets. This automation makes rule development accessible even to organizations without extensive ML expertise.
>
> **Precision where it matters most:** Importantly, there exist many safety-critical scenarios where other methods simply cannot provide the required guarantees. For example, consider a system that must achieve zero false positives in a specific domain for regulatory compliance, or a specialized assistant that must never discuss certain topics for legal liability reasons. GAVEL enables model owners to directly specify these constraints, while probabilistic classifiers and alignment-based methods offer only statistical assurances without hard guarantees.
>
> **Orthogonal and complementary:** We emphasize that GAVEL is not intended to replace existing safety mechanisms but to complement them. For many scenarios, general-purpose alignment or output moderation remains ideal. GAVEL specifically targets cases where explicit specification, auditability, and precision are paramount—addressing gaps that current methods cannot fill.
>
> We will expand the paper to include a more thorough discussion of when rule-based approaches are most appropriate versus when other safety mechanisms should be preferred, helping readers understand GAVEL's intended role within the broader safety landscape.
>
> Thank you for raising this concern—it has helped us clarify a crucial aspect of GAVEL's value proposition and practical deployment context.

---

> ### Author Response · Authors · 2025-11-21
> **Controlling TPR and FPR**
>
> > Is there a way to control the TPR and FPR?
>
> Excellent question\! Yes, GAVEL naturally supports flexible TPR-FPR tradeoffs through soft rule evaluation; instead of simply detecting if the CEs are present, we can multiply the CEs’ probabilities together and use the result as a confidence score for the rule’s presence.
>
> **Revision**: We will add ROC curves for each evaluation scenario, providing a sensitivity analysis showing the TPR-FPR tradeoffs achievable by varying this threshold. These demonstrate that GAVEL provides fine-grained control over operating points, allowing model owners to choose thresholds appropriate for their risk tolerance. For instance, a financial institution might set θ \= 0.85 to minimize false positives, while a content moderation system might use θ \= 0.65 to maximize harmful content detection.

---

> ### Author Response · Authors · 2025-11-21
> **Drop in Qwen3 Due to Multilingual Capacity**
>
> > Gemma and Qwen3 seem to have lower-weighted ACC in Figure 3\.
>
> This is a very interesting observation. We would like to clarify that Gemma does not have a lower weighted ACC overall; this drop is only truly visible in the "Racism" context. However, it is correct that Qwen3 performs worse across the board.
>
> We attribute these distinct behaviors to two different factors:
>
> 1. **Qwen3 (Consistent Drop):** We believe the "multilingual tax" hypothesis applies most strongly here. As a model designed for massive multilingual support, Qwen3's parameter space is distributed across many languages. This "capacity dilution" likely results in slightly less distinct activation signatures for English-specific safety concepts compared to English-centric models like Mistral or Llama 3\.
> 2. **Gemma (Specific Drop in Racism):** For Gemma-4B, the isolated performance drop in the Racism category is likely a result of **model size limiting semantic nuance**. The Racism evaluation is particularly challenging because it requires distinguishing between *hateful content* and *benign discussions about ethnic identity* (which we included in our negative datasets). While the 7B and 8B models successfully separated these overlapping concepts, the smaller 4B model appears to struggle with this finer sociological distinction in activation space, leading to the observed drop in accuracy.
>
> Practical implications: For production deployment of GAVEL on multilingual models, we recommend either training separate CE classifiers per language or using language-adaptive classifier architectures. We have added this recommendation to the Discussion section.
>
> Revision: We will update Section 4.2 to explicitly discuss these trade-offs regarding model size (Gemma) and multilingual capacity (Qwen).

---

> ### Comment · Reviewer_Hrih · 2025-11-24
>
> Thank you for the detailed response. My concerns are addressed, and thus move to a six.
>
> > Figure 2
>
> Your reasoning is understandable. I will leave it to the authors to decide on the placement. My preference is for an earlier release, as page 6 is pretty late in the paper, and possibly expanding the caption to include all the definitions might be helpful.
>
> > Drop In Qwen3 Due to Multilingual Capacity
>
> I think the figure should actually include the percentages because I believe it makes it difficult to compare the bars on such a small figure, which could lead to misinterpretation, as seen with my Gemma misunderstanding. On Qwen3, if you made the data to collect the activations in Chinese and English, then do you think this would increase the percentage on that plot (no experiment needed, just words is fine)?

---

> > ### Author Response · Authors · 2025-12-03
> > **Qwen3**
> >
> > We agree that expanding data collection to cover more languages would likely improve performance, particularly given Qwen's support for over 100 languages. Accordingly, we recommend that practitioners applying GAVEL to highly multilingual models curate native-language CE datasets to maximize detection accuracy.

---

> ### Author Response · Authors · 2025-12-04
> **Revision - Final Comment**
>
> Dear Reviewer, thank you very much for your time, thoughtful dialogue, and the opportunity you have given us to improve our work. We also sincerely appreciate your willingness to reconsider your initial assessment and raise your rating to a 6\. We hope that the additional experiments and revisions included in this final version further strengthen the contribution and meaningfully address the concerns you raised. We have completed all planned updates, and our detailed responses to each of your comments are provided below:
>
> * Clarity and Design Decisions
>   * We have carefully reviewed the paper to improve clarity and add any missing implementation details. We have fixed the issues with the notation and we have also moved the main framework figure up in the paper as per the reviewer’s recommendation. Moreover, we have added a discussion in the paper why we chose attention outputs over linear layer activations and we have added a complete ablation study to the **Appendix B** to support our insight as well.
> * GPT4 as baseline
>   * We addressed the missing comparison to a surface-level LLM moderator by adding a full evaluation to **Section 4.2.** There, we use a category-specific judge LLM (GPT-4) which was instructed explicitly on what to detect. The results are posted below and available in the paper: while GPT-4 performs comparably to GAVEL on clean, non-adversarial examples, it fails under even simple adversarial conditions. As shown in **Section 4.3** and **Appendix I**, GPT-4’s detection rate collapses under misdirection, goal hijacking, and evasion attacks (e.g., falling to 12% on Delusional Thinking and 15% on Election Interference), whereas GAVEL maintains \>85% recall across all categories because it monitors internal activations rather than surface text.
>   * We further highlight the need for activation based monitoring like GAVEL, by showing that **surface-level moderators cannot reliably detect when there is a hidden objective**. For this case, we made a new dataset on Deceptive Marketing: GPT-4 detects only 24% of violations, while GAVEL reaches 73% by identifying the co-activation of *Trust Seeding* and *Healthcare* concepts.
> * Rule Based Systems
>   * Regarding the subjectivity of rule-based systems, in the revised manuscript, we now address this point explicitly in the new **“Discussion on Limitations”** paragraph at the end of **Section 3.4**. There, we acknowledge that a Boolean, rule-based framework may initially seem restrictive, and we clarify that this explicitness is in fact a strength for high-precision, safety-critical settings: it enables model owners to clearly specify the internal states that count as violations, something current neural activation methods cannot provide. We also explain that scalability is preserved because CE and rule creation can be largely automated via our newly added agent-driven pipeline and supported through community sharing. This collaborative ecosystem directly mitigates concerns about subjectivity by allowing practitioners to choose and refine rules that match their own domain requirements, rather than relying on a single universal policy or misuse dataset. Finally, this section now makes it explicit that **GAVEL is orthogonal to other safety methods**, and can be layered alongside content moderation and alignment techniques to provide an additional high-precision safety layer where explicit guarantees are required.
>   * Additionally, earlier in **Section 3.4 (“Automating CE and Rule Development”)**, we highlight how GAVEL’s CE/rules can be automatically generated using an agentic tool over a shared CE/rule inventory. We provide an evaluation of this tool on data from the wild in **Section 4.2**  and provide further information on its design in the appendix. We are committed to open sourcing all of GAVELs code with the paper, including the automated tool to support practical scalability and cross-organizational reuse. In the meantime, the reviewer can explore the [live demo](https://gavelautomatedruleandcegenerationpipeline-b2ffsm7zwt6onjp2aago.streamlit.app/).
> * Controlling TPR and FPR
>   * In the revised paper, we now state explicitly (at the end of **Section 3.3**) that such control *is* supported, and we provide a full technical explanation and evaluation in **Appendix D (“ROC Analysis and Rule Scoring”)**. As described there, each rule is assigned a continuous confidence score computed as the geometric mean of the probabilities of its constituent CEs. This scoring function ensures that a rule’s score remains high only when *all* required CEs are activated, and it provides a smooth parameter for adjusting detection thresholds. Using these continuous scores, we generated full ROC curves for all misuse scenarios. The appendix reports that GAVEL achieves AUC values near 1.0 across almost all categories, demonstrating excellent separability and confirming that practitioners can reliably tune sensitivity to the desired TPR/FPR operating point.

---

> > ### Author Response · Authors · 2025-12-04
> >
> > We are truly grateful for the reviewer’s thoughtful and constructive review. The comments prompted several important clarifications and refinements that have made the paper both clearer and more impactful. In particular, the feedback helped us better articulate how activation-level rule systems can be implemented in practice and how to present our methods in a reproducible, transparent way. Thank you again for your time, insight, and engagement.

---

### Official Review · Reviewer_ihDH · 2025-10-30

**Soundness:** 2
**Presentation:** 2
**Contribution:** 2
**Rating:** 4
**Confidence:** 3

**Summary:**

This papers presents a novel framework for activation-based safety monitoring. In particular, the paper proposes to elicit cognitive elements from model activations, and then defining predicate rules over CEs and detects violations in real time using a multi-label classifier.  Finally, the proposed approach is evaluated on a self-generated datasets.

**Strengths:**

- The paper is to most extents clearly written and easy to follow
- Promising results on the self-generated benchmarks and informative an informative analysis in Figure 4.
- Informative Ablations (e.g. Transformer layer ablation and not just assuming some gold layer)

**Weaknesses:**

- **Boolean Predicate Rules:** While the grounding in concept space sounds intriguing, it remains unclear to me if the boolean logic is the right top-level tool. For instance, many nuance things are hard to express in boolean logic and it seems kind of a loss of flexibility (one of the core benefits of the current LLM-based paradigm)
- **Connection to SAEs:** Despite the paper's main motivation to build onto concept spaces, it seems odd to me that sparse autoencoders (SAEs) are not mentioned a single time. Especially as SAEs would offer "free" access to concept-level abstractions.
- **Interference of Concepts:** The authors state that "each activation set in the training collection H is curated to isolate a single CE a time", and list two advantages. However, the limitations of this approach remain open. For instance, how well can the resulting extraction deal with tokens where multiple concepts interfere?
- **Only Self-Generated Benchmarks:** While it's definitely a good evaluation tool to construct targeted datasets to test a proposed method, one should also evaluate the proposed method on existing benchmarks.
- **Baseline Comparisons:** Since you construct specific predicate rules for each dataset, did you also test a specialized prompt per category, and tested a vanilla model with this specialized prompt? How much does rule specialization to the specific dataset matter?
- **Lack of Details:** The paper lacks quite a lot of important deltas
  - Only very details on the RNN are given (would be hard to reproduce the paper)
  - No details on how the datasets were generated are given (also checked the appendix)

**Questions:**

**Additional Question (not related to weaknesses)**
- **New Adversarial Attack Surface:** While GAVEL defends against surface-level "representation attacks" , it introduces a new attack vector: the CE detector itself. A sophisticated adversary could, in principle, craft a "CE-level jailbreak"—a prompt that achieves a harmful outcome without triggering the specific activation signatures the multi-label classifier is trained to detect. This vulnerability is not explored in the paper.

I am more than happy to raise my score if the authors can address my concerns and questions, and in particular make the paper "complete" such that it could be reproduced. In the current state, it would be hard to reproduce.

---

> ### Author Response · Authors · 2025-11-12
> **baseline comparisons - clarification question**
>
> > Baseline Comparisons: Since you construct specific predicate rules for each dataset, did you also test a specialized prompt per category, and tested a vanilla model with this specialized prompt? How much does rule specialization to the specific dataset matter?
>
> Thank you for the comment. Could you please clarify what you mean by “a vanilla model with this specialized prompt”? Specifically, are you suggesting:
>
> (a) prompting a base LLM (without any activation-level monitoring) to detect or classify examples in each dataset using a category-specific textual prompt (e.g., “Identify if this is phishing content”), or
>
> (b) using a baseline activation-based model with dataset-specific training, but without our rule-based composition?
>
> Or perhaps something else?
> We want to ensure we interpret your comment correctly before responding

---

> > ### Comment · Reviewer_ihDH · 2025-11-18
> > **clarification**
> >
> > I meant **prompting a base LLM (without any activation-level monitoring) with a category-specific textual prompt (including the concepts of your predicate logic)**. IMO, this would serve as a fair and possibly stronger alternative baseline to the moderation models.

---

> ### Author Response · Authors · 2025-11-21
> **Boolean Predicate Rules**
>
> > **Boolean Predicate Rules:** While the grounding in concept space sounds intriguing, it remains unclear to me if the boolean logic is the right top-level tool. For instance, many nuance things are hard to express in boolean logic and it seems kind of a loss of flexibility (one of the core benefits of the current LLM-based paradigm)
>
> Thank you for the thoughtful feedback. Although GAVEL is based upon boolean logic, we believe that it is possible to achieve sufficient nuance in rule composition by leveraging LLMs. We also view the boolean nature of GAVEL as a strength for when precision and well-defined constraints are necessary. We will elaborate on each:
>
> 1. **Flexible Rule Composition:**
> The process of composing rules and defining CEs can be done automatically using an LLM agent. This enables us to leave the challenging nuance of defining rules and their supporting CEs to language models. To demonstrate this, we have built and published a tool which you can try out now: [Gavel Automatation Tool](https://gavelautomatedruleandcegenerationpipeline-b2ffsm7zwt6onjp2aago.streamlit.app/)
> Given a user description of the detection needs, the tool uses an agentic pipeline to automatically define new rules, CEs, and their respective excitation datasets. With this, all a user needs to do is train the multilabel classifier on the activations from these datasets. Moreover, if a user provides a database of existing rules and CEs (potentially shared by the GAVEL community online) then the tool will utilise this when reasoning over what additional rules and CEs are needed (to minimize redundant work and build off the work of others).
>
> 2. **Necessity of Boolean Rule-based Precision:**
> Some applications require very high precision and extremely low false-positive rates. Current activation-based safeguards offer no way for practitioners to *explicitly* define the states that count as a violation. Moreover, current moderation tools struggle to identify these explicit cases in the presence of representation and prompt-injection attacks.
> GAVEL addresses this by allowing users to specify the internal states directly through Boolean predicates. Like other rule-based cybersecurity systems (e.g., Snort, YARA), GAVEL’s use of boolean predicates enable users to not only enforce clear and well-defined rules, but also enable users to share new rules (e.g., malicious signatures) across different ecosystems.
>
> In situations where Boolean rules might feel restrictive, it’s important to note that GAVEL is not exclusive. It can be layered with text-level moderation, classifier-based tools, or other safety mechanisms. Those tools provide broader, more abstract safety coverage, while GAVEL enforces the precise, policy-critical behaviors where accuracy matters most \-just as cybersecurity combines multiple layers (firewalls, IDS, malware scanning) to harden a system.
>
> Finally, the ability to precisely define the states of interest opens a path for AI governance. A clearly defined, model-agnostic rule set allows organizations and policymakers to adopt standardized safety requirements and apply them consistently across many models, enabling transparent and enforceable safety practices.
>
> **Revision**: We will revise the paper to clarify how GAVEL can be made more flexible through agentic, LLM-assisted rule composition \-and we will open-source our rule-generation tool to support this. We will also expand our discussion of the advantages and limitations of rule-based systems, and explain how GAVEL can also complement other safety approaches.

---

> ### Author Response · Authors · 2025-11-21
> **Connection to SAEs**
>
> > **Connection to SAEs:** Despite the paper's main motivation to build onto concept spaces, it seems odd to me that sparse autoencoders (SAEs) are not mentioned a single time. Especially as SAEs would offer "free" access to concept-level abstractions.
>
> We thank the reviewer for this excellent observation. We agree that Sparse Autoencoders (SAEs) are a vital development in interpretability and resolving superposition. While SAEs offer a powerful bottom-up method for feature discovery, we chose a supervised classification approach for GAVEL to address three practical realities of safety governance:
>
> * **Direct Definition vs. Curated Search:** The core challenge in safety governance is ensuring the detector strictly aligns with a specific policy definition (e.g., "UK Tax Fraud" vs. generic "Deception"). While SAEs excel at disentangling features, and automated interpretation techniques (like auto-labeling) are improving, using SAEs remains a "search" process—practitioners must hope the SAE has learned a feature that aligns with their specific policy nuance and then locate it. *GAVEL reverses this workflow*: it uses a "top-down" approach where model owners explicitly *define* the target concept via the excitation dataset. This ensures the detector represents the exact behavior proscribed by the policy without requiring a post-hoc search of the latent space.
> * **Cross-Organizational Portability:** A central goal of GAVEL is to enable community-shared rulesets akin to cybersecurity signatures. SAE features are tied to specific model weights; a feature index in Llama-3 does not map to Mistral-7B. In contrast, GAVEL relies on textual excitation datasets which are model-agnostic. This allows safety rules to be ported instantly across different organizations and architectures, whereas an SAE approach would require independently training and mapping dictionaries for every model instance.
> * **Computational Efficiency:** The reviewer notes that SAEs offer "free" access to abstractions. While accessing features from a *pre-trained* SAE is efficient, training the SAE itself remains computationally expensive (often requiring billions of tokens). GAVEL’s targeted probes are trained on small, specific datasets (\~150 examples), making them orders of magnitude cheaper to set up for specific safety needs
>
> **Future Integration:** Crucially, we view GAVEL as a modular framework. The "Cognitive Element" is an abstraction layer. Today, we implement this via supervised probes for efficiency. However, as SAE automated mapping matures and becomes more standard, SAE features could easily "slot in" as the underlying detection mechanism for a Cognitive Element, without requiring changes to the logic or rule structure of the GAVEL framework.
>
> **Revision:**  Based on this feedback, we will make the following changes to the paper:
> * **Update Related Work (Section 2):** We will add a dedicated discussion on Sparse Autoencoders, acknowledging their ability to extract monosemantic features and contrasting their "bottom-up" discovery nature with GAVEL’s "top-down" definition approach.
> * **Update Future Work (Section 5):** We will explicitly state that future iterations of GAVEL could leverage SAEs as an alternative backend for detecting Cognitive Elements, particularly as automated semantic mapping tools improve.

---

> ### Author Response · Authors · 2025-11-21
> **Interference of Concepts**
>
> > **Interference of Concepts:** The authors state that "each activation set in the training collection H is curated to isolate a single CE a time", and list two advantages. However, the limitations of this approach remain open. For instance, how well can the resulting extraction deal with tokens where multiple concepts interfere?
>
> Thank you for noting this, we forgot to support this claim in the paper. In our experiments, we found that the multilabel classifier can indeed detect multiple CEs at the same time.
>
> **Revision**: We will add to the appendix an evaluation that shows that CEs can co-occur without conflict. In the meantime, we invite the reviewer to explore our GAVEL visualization tool, which illustrates these co-occurrences interactively: [GAVEL Interactive Demo](https://44vpkxck01df4f-8501.proxy.runpod.net/)
> We will publish this visualization tool with the paper as well.

---

> ### Author Response · Authors · 2025-11-21
> **Only Self-Generated Benchmarks**
>
> > **Only Self-Generated Benchmarks:** While it's definitely a good evaluation tool to construct targeted datasets to test a proposed method, one should also evaluate the proposed method on existing benchmarks.
>
> Thank you for the insight! We generated our datasets because there weren't many benchmarks that fit our criteria: many samples of one specific misuse/rule (compared to a generic misuse category). However, the paper would certainly benefit from using existing benchmarks as well.
>
> **Revision:** We will evaluate our rules on several benchmark datasets, and update you when the revised paper is ready.

---

> ### Author Response · Authors · 2025-11-21
> **Baseline Comparisons**
>
> >**Baseline Comparisons:** Since you construct specific predicate rules for each dataset, did you also test a specialized prompt per category, and tested a vanilla model with this specialized prompt? How much does rule specialization to the specific dataset matter?
>
> Thank you for this comment. While a judge LLM can succeed quite well at detecting our scenarios \- they are easy to evade because the LLM relies purely on raw text. The surface-level moderation can be bypassed. For example:
>
> 1. **Misdirection**: shifting an LLM’s focus away from the relevant content by adding benign content.
> 2. **Hijacking**: the use of prompt injection to override the system instructions
> 3. **Evasion**: by telling the generating LLM that it’s being monitored and that it must “achieve the goal without being detected”.
>
> In contrast, GAVEL operates on activations, so we can still detect the underlying Cognitive Elements as they arise—regardless of how the generating model phrases content or plants LLM-based traps.
>
> There are also many scenarios where the harmful motive never appears explicitly in the raw text at all. For example, an LLM that is prompted to provide all future summaries with a systematic bias, perform deceptive marketing over a phone call, or subtly steer a user’s decisions. These behaviors show up in internal representations long before (or even without) appearing in the text, which is exactly why activation-level monitoring is needed.
>
> In short: text-only moderation sees *what the model says*, but GAVEL reveals *what the model is actually doing*.
>
> **Revision**: We have run the experiments and present the results here:
>
> ### Baseline
>
> | Category | TPR (GAVEL) | TPR (GPT4) | TPR (GPT4 with no CEs/Rules) | FPR (GAVEL) | FPR (GPT4) | FPR (GPT4 with no CEs/Rules) |
> | :---- | ----- | ----- | ----- | ----- | ----- | ----- |
> | Phishing  | 0.95 | **0.98** | 0.91 | **0.00** | **0.00** | **0.00** |
> | SQL Injection | 0.89 | 0.99 | 1.00 | **0.00** | 0.09 | **0.00** |
> | Delusional  | **0.90** | **0.90** | 0.84 | 0.01 | **0.00** | 0.01 |
> | Anti-LGBTQ | **1.00** | 0.97 | 0.99 | **0.00** | 0.03 | **0.00** |
> | Elections | **0.99** | 0.85 | 0.66 | 0.01 | 0.30 | **0.00** |
> | Racism | **0.99** | 0.97 | **0.99** | 0.02 | 0.05 | **0.00** |
> | Tax Authority | 0.86 | 0.96 | 1.00 | **0.00** | 0.02 | **0.00** |
> | Romance | **1.00** | 0.95 | **1.00** | 0.01 | **0.00** | **0.00** |
> | e-commerce | **0.90** | 0.87 | 0.87 | 0.04 | 0.02 | **0.00** |
>
> ### Attacks against LLM Monitor (TPR)
>
> | Task | Misdirection (GAVEL) | Misdirection (GPT4) | Goal Hijacking (GAVEL) | Goal Hijacking (GPT4) | Evasion (GAVEL) | Evasion (GPT4) |
> | :---- | ----- | ----- | ----- | ----- | ----- | ----- |
> | Phishing | **0.97** | 0.55 | **0.96** | 0.55 | **1.00** | 0.89 |
> | SQL Injection | **0.89** | 0.49 | **0.89** | 0.70 | **0.94** | 0.85 |
> | Delusional | **0.87** | 0.12 | **0.87** | 0.49 | **0.91** | 0.44 |
> | Anti-LGBTQ | **1.00** | 0.35 | **1.00** | 0.90 | **0.99** | 0.13 |
> | Elections | **0.99** | 0.15 | **0.99** | 0.63 | **1.00** | 0.49 |
> | Racism | **0.99** | 0.16 | **0.99** | 0.91 | **0.98** | 0.57 |
> | Tax Authority | **0.86** | 0.00 | **0.86** | 0.28 | 0.73 | **0.83** |
> | Romance | **1.00** | 0.24 | **1.00** | 0.48 | **0.95** | 0.74 |
> | e-commerce | **0.89** | 0.01 | **0.90** | 0.12 | **0.99** | 0.92 |
>
> We see that GAVEL is not affected by the attacks however the TPR of the GPT monitor plummets under simple attacks.

---

> ### Author Response · Authors · 2025-11-21
> **Lack of Details**
>
> >**Lack of Details:** The paper lacks quite a lot of important details.
> >* Only very details on the RNN are given (would be hard to reproduce the paper).
> >* No details on how the datasets were generated are given (also checked the appendix)
>
>
> We sincerely apologize for these omissions. Reproducibility is paramount, and we will add all of these missing details in the revised paper (we will notify you when it’s ready).
> To further support reproducibility, we will not only publish the entire GAVEL code base on GitHub but also the two supporting tools ([GAVEL Automation Tool](https://gavelautomatedruleandcegenerationpipeline-b2ffsm7zwt6onjp2aago.streamlit.app/)  and [GAVEL Interactive Demo](https://44vpkxck01df4f-8501.proxy.runpod.net/)).
>
> We appreciate the reviewer's emphasis on reproducibility and believe these additions make our work substantially more accessible to the community.

---

> ### Author Response · Authors · 2025-11-21
> **New Adversarial Attack Surface**
>
> >**New Adversarial Attack Surface:** While GAVEL defends against surface-level "representation attacks" , it introduces a new attack vector: the CE detector itself. A sophisticated adversary could, in principle, craft a "CE-level jailbreak"—a prompt that achieves a harmful outcome without triggering the specific activation signatures the multi-label classifier is trained to detect. This vulnerability is not explored in the paper.
>
> This is a wonderful insight! We agree that, as a safety paper, the paper should highlight this new attack surface.
> In principle, an adaptive adversary could attempt to evade a rule by suppressing one of its required CEs. However, this may be far more difficult in practice. Tactics such as rephrasing do not meaningfully change the model’s latent semantic activations. To suppress a CE, an adversary must alter the underlying activation pattern itself, which typically requires changing the semantic content of the attack. This may limit the utility of the attack -putting the defender at an advantage. For example, in an IRS scam, applying pressure requires generating threats. If the adversary tries to avoid the Threat CE, the scam becomes less effective. However, there may be other forms of token-based adversarial examples that should be studied.
>
> **Revision:** We will add this discussion to the paper and call upon the community to further research this new attack surface to make the novel domain of rule-based activation safety a robust option.

---

> ### Author Response · Authors · 2025-12-04
> **Revision - Final Comment**
>
> Dear Reviewer, thank you very much for your time, constructive feedback, and thoughtful consideration of our work. First, we would like to note that we have worked hard to make sure all of the details necessary to reproduce our work are now in the paper. We are also committed to release all of our source code (for GAVEL and its new automated CE/rule generation pipeline) because we strongly believe in the positive impact GAVEL can have in a community setting.
>
> We completed all of the planned revisions and addressed each of your comments as outlined below:
>
> * Boolean Predicate Rules
>   * The new version of the paper resolves the issue of flexibility and scalability of our rule-based system in two ways. First, we have added a subsection to (**Section 3.4 “Automating CE and Rule Development”**) that explains how rule development and CE creation can be fully  automated using an agentic pipeline. We also detail how the pipeline works **(in the Appendix E)** and explain that it can use an existing community CE/rule inventory to build on the work of others, as shown in the [live demo](https://gavelautomatedruleandcegenerationpipeline-b2ffsm7zwt6onjp2aago.streamlit.app/). Then in the evaluation section **(Section 4.2)**, we test the pipeline on new benchmark datasets taken from the wild  (PKU-SafeRLHF phishing, ReasoningShield political risk, and ToxiGen racism) to demonstrate its effectiveness: **without any manual intervention** the tool developed the rules and training data for the classifier which then achieved strong “out-of-the-box” detection performance (e.g., TPR ≈ 0.97 on political risk and 0.91 on ToxiGen). We commit to releasing the full GAVEL source code, including the automation pipeline, upon publication.
>   * Second, we have added a discussion to **Section 3.4** that explains how collaboration via CE/rule sharing in the community will reduce the challenge of mapping out all relevant misuse settings. We then support this echoing how this is done today in cyber security in similar rule-based systems. There we also note the limitations of GAVEL and how it can be used in parallel to other safety systems as a means to enhance precision.
> * Connection to SAEs
>   * We have revised the **related work section** to talk about SAEs by acknowledging their ability to extract monosemantic features and contrasting their "bottom-up" discovery nature with GAVEL’s "top-down" definition approach. We have also made mention of how SAEs could be used to potentially replace CEs in GAVEL as future work in our **conclusion**.
> * Interference of Concepts
>   * We have revised the paper to explicitly state that the multi-label classifier can indeed detect multiple CEs on a single token, even though it was trained on each CE independently. In the **Appendix C**, we expand on this claim by providing both an evaluation and visual examples from our datasets: we found that 54% of all samples in our datasets included the event of co-occurrence of CEs. This finding aligns with expectations, since the 23 CEs capture distinct semantic behaviors.
> * Only Self-Generated Benchmarks
>   * As mentioned above, to address the concern regarding a lack of evaluation on public benchmarks, the revised paper now includes an assessment of three established safety datasets: PKU-SafeRLHF (phishing), ReasoningShield (political risk), and ToxiGen (hate speech). Using our automated CE/rule generation pipeline, we used GAVEL to detect these cases **without any manual intervention** \-by only providing the tool with a natural language description of the setting. The results presented in **Section 4.2** with its full details in the **Appendix H**. demonstrate that GAVEL generalizes effectively to real, diverse public datasets.
> * Baseline Comparisons
>   * We addressed the missing comparison to a surface-level LLM moderator by adding a full evaluation to **Section 4.2.** There, we use a category-specific judge LLM (GPT-4) which was instructed explicitly on what to detect. The results are posted below and available in the paper: while GPT-4 performs comparably to GAVEL on clean, non-adversarial examples, it fails under even simple adversarial conditions. As shown in **Section 4.3** and **Appendix I**, GPT-4’s detection rate collapses under misdirection, goal hijacking, and evasion attacks (e.g., falling to 12% on Delusional Thinking and 15% on Election Interference), whereas GAVEL maintains \>85% recall across all categories because it monitors internal activations rather than surface text.
>   * We further highlight the need for activation based monitoring like GAVEL, by showing that **surface-level moderators cannot reliably detect when there is a hidden objective**. For this case, we made a new dataset on Deceptive Marketing: GPT-4 detects only 24% of violations, while GAVEL reaches 73% by identifying the co-activation of *Trust Seeding* and *Healthcare* concepts.

---

> > ### Author Response · Authors · 2025-12-04
> >
> > * Lack of Details
> >   * We have updated Section 4.1 to provide a complete specification of our experimental setup to ensure reproducibility. We now explicitly detail the detector architecture (a multi-label RNN with three GRU layers and 256 hidden units), the input processing (5-token segments with a stride of 5), and the specific training hyperparameters (300 samples per CE, 80:20 train-test split, Adam optimizer with a learning rate of 3e\-4, and Binary Cross Entropy loss). Additionally, we confirm the use of GPT-4.1 for data generation and reiterate our commitment to releasing the full codebase to reproduce all experiments and datasets with the paper.
> > * New Adversarial Attack Surface
> >   * In the revised manuscript, we have added a dedicated discussion of this new attack surface to the **end of Section 4.2**. There, we explicitly describe the notion of a *CE-level jailbreak* \-an attack that attempts to elicit harmful behavior without activating the corresponding CE signatures. As reflected in the revised text, we note that such attacks are inherently difficult to execute because suppressing the relevant CE typically requires weakening the harmful behavior itself (e.g., conducting a tax scam without triggering threat-related activations). Nonetheless, we agree this represents an important emerging threat model, and the revised manuscript now calls for future work to investigate this domain and further strengthen rule-based activation safety.
> >
> > We sincerely appreciate the reviewer’s insightful feedback, which helped us not only ensure reproducibility of our work, but also ensure that the idea of a rule-based system on activations  can be a practical and meaningful mechanism in AI safety. The reviewer’s comments have directly strengthened the paper and improved its value to the broader AI safety community. Thank you\!

---

### Official Review · Reviewer_VVtu · 2025-11-01

**Soundness:** 3
**Presentation:** 3
**Contribution:** 3
**Rating:** 6
**Confidence:** 2

**Summary:**

The paper introduces GAVEL, a framework for improving LLMs safety by monitoring internal activations. It proposes the use of Cognitive Elements, as well as logical rules to detect and respond to unsafe or policy-violating behaviors. The framework is inspired by rule-sharing practices in cybersecurity and aims to foster community collaboration in defining and maintaining safety standards.

**Strengths:**

- The use of Cognitive Elements allows for composable safety rules that are explainable
- GAVEL operates with minimal computational overhead
- The framework has good performance across different LLM architectures and languages

**Weaknesses:**

- The process of defining Cognitive Elements and composing rules is currently manual and may not scale easily

**Questions:**

- Are there avenues/ possibilities for automating the rule creation?

---

> ### Author Response · Authors · 2025-11-21
> **Automating Rule and Cognitive Element (CE) Generation**
>
> > The process of defining Cognitive Elements and composing rules is currently manual and may not scale easily. Are there avenues / possibilities for automating the rule creation?
>
> We appreciate this important question about practical scalability. We believe that GAVEL is highly practical and scalable for two reasons:
>
> 1. **Automation**: The process of composing rules and defining CEs can be done automatically using an LLM agent. To demonstrate this, we have built and published a tool which you can try out now: [Gavel Automatation Tool](https://gavelautomatedruleandcegenerationpipeline-b2ffsm7zwt6onjp2aago.streamlit.app/)
> Given a user description of the detection needs, the tool uses an agentic pipeline to automatically define new rules, CEs, and their respective excitation datasets. With this, all a user needs to do is train the multilabel classifier on the activations from these datasets. Moreover, if a user provides a database of existing rules and CEs (potentially shared by the GAVEL community online) then the tool will utilise this when reasoning over what additional rules and CEs are needed (to minimize redundant work and build off the work of others).
>
> 2. **Community Collaboration**: We strongly believe that a strength and advantage of GAVEL is that it *is* a rule-based system. Rule-based systems in the cyber security domain thrive and scale because they rely on shared development and reuse. Like malware signatures or intrusion and firewall rules, GAVEL is designed for cross-vendor collaboration: once a Cognitive Element (CE) or rule is created and validated, it can be easily shared and applied by others. We envision a public repository of shared tested CEs and rules to emerge. This lowers the barrier to entry for new users who are simply seeking a plug and play solution. For example, if one vendor identifies a new LLM-powered social-media scam, they can contribute the corresponding rules and the entire community benefits.
>
> **Revision**: We will revise the paper in the coming weeks to include a discussion of how rule composition can be scaled via automation. We will also publish our automation pipeline’s source code with this publication.

---

> ### Author Response · Authors · 2025-12-04
> **Revision - Final Response**
>
> Dear reviewer, thank you so much for your time and consideration. We have completed the revisions as planned according to each item:
>
> #### **Automating Rule and Cognitive Element (CE) Generation**
>
> The new version of the paper resolves the issue of scalability in two ways. First, we have added a subsection to (**Section 3.4 “Automating CE and Rule Development”**) that explains how rule development and CE creation can be fully  automated using an agentic pipeline. We also detail how the pipeline works **(in the Appendix E)** and explain that it can use an existing community CE/rule inventory to build on the work of others, as shown in the [live demo](https://gavelautomatedruleandcegenerationpipeline-b2ffsm7zwt6onjp2aago.streamlit.app/). Then in the evaluation section **(Section 4.2)**, we test the pipeline on new benchmark datasets taken from the wild  (PKU-SafeRLHF phishing guidance, ReasoningShield political risk, and ToxiGen hatespeech) to demonstrate its effectiveness: **without any manual intervention** the tool developed the rules and training data for the classifier which then achieved strong “out-of-the-box” detection performance (e.g., TPR ≈ 0.97 on political risk and 0.94 on ToxiGen). We commit to releasing the full GAVEL source code, including the automation pipeline, upon publication.
>
> Second, we have added a discussion to **Section 3.4** that explains how collaboration via CE/rule sharing in the community will reduce the challenge of mapping out all relevant misuse settings. We then support this echoing how this is done today in cyber security in similar rule-based systems.
>
> We are grateful to the reviewer for prompting us to clarify and substantiate how CE and rule creation can scale in practice. These changes have truly made our paper significantly more useful to the AI Safety community in its present state.

---

### Author Response · Authors · 2025-11-12
**Author Comment: Initial Response and Plan for Revisions**

Thank you for your thoughtful and constructive review of "GAVEL: Towards Rule-Based Safety through Activation Monitoring." We genuinely appreciate the time you invested in our work and the helpful feedback you provided.

We are actively working through the points you raised. Rather than waiting to address everything at once, we plan to post responses incrementally as we complete each set of experiments and analyses. This will allow us to share results with you more promptly while maintaining the rigor and detail each point deserves.

We look forward to engaging with your feedback and will begin posting updates soon.

---

### Author Response · Authors · 2025-11-21
**Authors Comment - General**

We are deeply grateful to all reviewers for their thorough evaluation and insightful feedback. Your patience and constructive criticism have been invaluable in strengthening this work. We have carefully addressed each comment raised, and our detailed responses are provided below.

**Status of revisions**: We are currently completing additional experiments requested by reviewers, including extended benchmark evaluations and robustness analyses. Once these experiments conclude, we will promptly update the reviewers with the complete results and the revised manuscript on OpenReview.

**Addressing Scalability and Interpretability**: In response to reviewer concerns about scalability and interpretability, we are pleased to announce two interactive demonstrations that substantiate GAVEL's practical viability:

1. **Regarding scalability of the rule-based system**: We are excited to let the reviewers know that we have published an online app that automates the entire process of rule composition and CE dataset generation using LLM agents. The tool demonstrates that GAVEL can be set up in a very quick, and automated manner minimizing any manual effort required. The reviewers can access this demo here: [Gavel Automatation Tool](https://gavelautomatedruleandcegenerationpipeline-b2ffsm7zwt6onjp2aago.streamlit.app/)
   The reviewers are encouraged to experiment with their own safety scenarios to observe the automation capabilities firsthand.
2. **Interactive GAVEL Visualization Interface:** To address questions about interpretability and how human auditors interact with GAVEL in practice, we have deployed a live demonstration running GAVEL on Mistral. This interface allows reviewers to input arbitrary prompts, observe real-time CE activations during generation, visualize which rules are triggered and why, and explore the system's decision-making process interactively.
   Access the demo here: [GAVEL Interactive Demo](https://44vpkxck01df4f-8501.proxy.runpod.net/)

**Code release commitment:** The complete source code for both tools (above), along with the full GAVEL framework implementation, all trained models, evaluation scripts, and datasets, will be released as a public GitHub repository upon paper acceptance. We are committed to full reproducibility and community adoption.

We look forward to your feedback and remain available for any clarifications you may request.

The Authors

---

### Author Response · Authors · 2025-12-04
**Author Note to AC**

Dear Area Chair,

Thank you very much for your time and consideration of our submission. We greatly appreciate both your efforts and the reviewers’ positive and constructive feedback, as well as the care everyone has taken in engaging with our work. Here we provide a brief summary of how we have addressed the reviewers’ concerns, along with the relevant context from the discussion-phase feedback (still visible below).

**Addressing the reviewer’s main concerns: scalability and GPT-4 comparison.**
 Across the reviews, two common concerns emerged:

1. **Scalability of a rule-based system and effort of manual CE/rule creation.**
   *(VVtu, x4Ks)*
   We have resolved this concern in two ways:
   * We explain in the revised paper that the entire process of creating rules, their CEs, and their respective excitation datasets, can be fully automated using an LLM-driven agentic pipeline. To support the claim, ensure reproducibility, and enable adoption of this approach, we have:
     * Published a fully functional interactive application that automates the entire process (we invite the AC to try it out [here](https://gavelautomatedruleandcegenerationpipeline-b2ffsm7zwt6onjp2aago.streamlit.app/)). The source code will be made open source on GitHub after acceptance.
     * Provided complete implementation details in the paper; a description in the body of the paper and complete details in the appendix.
     * Evaluated the performance of the rules generated by the pipeline on additional test sets and scenarios (available in the Appendix H).
   * We clarify in the paper why rule-based safety is both practical and scalable by drawing on the long-standing success of rule sharing in cybersecurity. In that field, community-maintained signatures, intrusion rules, and policy definitions allow organizations to collaboratively strengthen one another’s defenses, achieving broader coverage and faster adaptation than any individual team could manage alone. GAVEL follows this model. Once a CE or rule is defined and validated, it can be reused, adapted, and combined by others, enabling a growing shared vocabulary of safety behaviors. This collaborative dynamic makes rule-based activation monitoring not only feasible at scale but increasingly powerful as the community contributes additional rules over time. Especially since **GAVEL is model agnostic**.

2. **Comparison to text-only baselines (GPT4).**
   *(Hrih, ihDH, x4Ks)*
   * **Request**. The reviewers asked for (1) a comparison to a “vanilla” LLM (specifically GPT4) with specialized prompts and for (2) an evaluation of GAVEL’s robustness under adversarial prompting. In the revision, we added a GPT-4 baseline across all categories and evaluated both systems under three adversarial conditions: misdirection, prompt injection, and evasive conversation. We also evaluate a new “deceptive marketing” category designed to test the detection of models given hidden objectives.
   * **Results.** GPT-4 performs well on clean data, but its detection rates collapse under all adversarial conditions, while GAVEL remains consistently robust. Because GAVEL analyzes a model’s internal activations, it is resistant to representation-level attacks and prompt injection. In scenarios involving hidden objectives (e.g., deceptive marketing), GPT-4 almost always fails, demonstrating that surface-level judges cannot reliably infer intent from text alone, whereas activation-based methods can.
   * **Conclusion.** Surface-level moderators like GPT-4 can be effective on straightforward inputs but are vulnerable to a broad range of evasion tactics and **cannot detect hidden objectives**. These experiments directly address whether a strong text-only judge could replace activation-level monitoring, and they show that activation-based approaches like GAVEL provide essential gains in robustness and precision.

In addition to these main concerns, we have addressed **all other reviewer comments** in the revised manuscript, including: clearer architectural justifications (e.g., attention outputs vs. linear layers, RNN choice **with full ablation studies** to support them), expanded details for reproducibility, discussion of connections to SAEs, treatment of CE co-occurrence, and TPR/FPR operating points. For details, we respectfully refer the AC to our point-by-point responses to each reviewer below.

---

> ### Author Response · Authors · 2025-12-04
>
> **Context from the discussion phase and ratings.**
> We would also like to draw the AC’s attention to several specific comments from the discussion phase (still visible below). Although we had not yet presented our new evaluation results, the reviewers positively welcomed our initial round of responses. Unfortunately, their updated numerical ratings are no longer reflected due to the recent OpenReview incident:
>
> * **Reviewer Hrih (original score 2\)** explicitly wrote:
>    “Thank you for the detailed response. My concerns are addressed, and thus move to a six.”
>   This increase occurred **before** we added the final experimental results the reviewer requested (e.g., additional ablations and clarifications).
>
> * **Reviewer x4Ks (original score 8\)** similarly wrote:
>    “Sounds good. Changed Ratings.”
>   Indicating an improved overall assessment following our clarifications, again before the additional experiments and revisions were posted or revised in the paper.
>
> * **Reviewer ihDH  (original score 4\)** wrote:
>    “I am more than happy to raise my score if the authors can address my concerns and questions, and in particular make the paper ‘complete’ such that it could be reproduced.”
>    In the revised manuscript and responses, we have **done exactly what was requested**:
>   * We have published the automation and visualization applications as well as provided all of the implementation details in the paper and appendix. We will also publish all of the source code on GitHub, including the automation applications in the camera-ready. Moreover, we addressed the issue of scalability, introduced the GPT-4 baselines and adversarial evaluations, provided an evaluation on CE interference, added public benchmark datasets and added an expanded discussion on SAEs.
>
> Taken together, the discussion-phase comments and our revisions on **scalability**, **GPT-4 comparison**, and **reproducibility** were addressed,
> In addition, we have even updated our main results table with ten times more test samples across all baselines to provide a more accurate perspective. We are confident that, together with the discussion phase comments, and our revisions by addressing the scalability and surface-level detectors, have resolved the main issues raised by all reviewers.
>
> **Closing remarks.**
> At present, there is **no established rule-based framework for AI safety operating directly on neural activations**. Yet such a framework is urgently needed for:
>
> * **Higher precision and lower false positives**,
> * **Interpretability and auditability** (human-understandable CEs and rules), and
> * **Standardization of safety requirements;** rulesets to support legislation, regulation, and broader AI governance.
>
> GAVEL introduces **rule-based safety for LLMs at the activation level**, bringing a well-established defense paradigm from cybersecurity—rule and signature sharing—into modern AI systems. It is designed to foster an ecosystem of **community cooperation**, where organizations can share, adapt, and extend CEs and rules, much like intrusion detection and malware signatures today.
>
> We believe this paper not only opens a **new domain of activation-based, rule-driven AI safety**, but also has **direct, practical impact** on how safety can be engineered and governed in current LLM deployments. For additional context and to see GAVEL in action, we invite the AC to explore our public, anonymous GAVEL tools (the [**Automation Tool**](https://gavelautomatedruleandcegenerationpipeline-b2ffsm7zwt6onjp2aago.streamlit.app/) and [**Interactive Demo**](https://44vpkxck01df4f-8501.proxy.runpod.net/)) .
>
> Thank you again for your time and consideration. We recognize the significant time and effort required in your role, and we truly appreciate your careful attention to our work.
>
> The Authors

---

### Meta-Review · Area_Chair_9iAT · 2026-01-07

**Summary:**

The paper proposes GAVEL, a rule-based activation monitoring framework for LLM safety. Reviewers appreciate its originality, clarity, and significance. The major concerns include scalability, the lack of evaluation on existing benchmarks, insufficient methodological details, writing issues, limited generalizability, and insufficient depth of validation.

**Reviewer Concerns:**

Reviewer concerns addressed by the rebuttal:
1. The scalability issue is addressed through an LLM-driven agentic pipeline. (Reviewers VVtu, x4Ks)
2. A GPT-4 baseline has been added. (Reviewers Hrih, ihDH, x4Ks)
3. Additional reproducibility details have been provided. (Reviewer ihDH)


Reviewer concerns that may still be outstanding:
1. The concern that the CE detector introduces a new attack surface may still be under-explored. (Reviewer ihDH)
2. The concern about the expressiveness of Boolean rules may still remain. (Reviewer ihDH)

**Reviewer Scores:**

Reviewer VVtu is likely to maintain a score of 6. Reviewer ihDH is likely to maintain a score of 4. Reviewer Hrih is likely to raise their score to 6 as stated. Reviewer x4Ks is likely to maintain a score of 8.

---

### Decision · Program_Chairs · 2026-01-26

Accept (Poster)